# Position: State-of-the-Art Claims Require State-of-the-Art Evidence

**YongKyung Oh** [1]

## Abstract

State-of-the-Art (SOTA) claims pervade Artificial Intelligence (AI) and Machine Learning (ML) research. These claims rest on benchmark evaluations, where models are ranked by aggregate scores across tasks. Public benchmarks or leaderboards are the most visible instance, but the same structure appears in paper tables throughout the literature. However, such minimal evidence often cannot support these strong claims. We identify a widespread claim-evidence gap in AI benchmarking. Claiming SOTA carries implicit assumptions beyond mean score superiority, suggesting that a model meaningfully outperforms alternatives across most tasks. However, a marginal improvement in the mean score merely indicates a top average rank rather than true superiority. Analyzing ten cross-domain benchmarks from public leaderboards, we found that in more than half of top-model comparisons, at least one commonly assumed property of superiority does not hold. These properties include meaningful effect size, consistency across tasks, or robustness to dataset removal. Instead, aggregate gains are frequently driven by outlier datasets. This fragility persists even in benchmarks with many tasks. We argue that claim language should reflect the strength of the underlying evidence. This requires no additional experiments, only honest reporting of what results actually show, enabling more precise and interpretable comparisons across models.

## 1. Introduction

State-of-the-Art (SOTA) claims shape how Artificial Intelligence (AI) and Machine Learning (ML) research measures progress. Models are compared across benchmark tasks, and the top performer by aggregate score is declared superior.

---

[1]Medical & Imaging Informatics (MII), University of California, Los Angeles (UCLA). Correspondence to: YongKyung Oh <yongkyungoh@mednet.ucla.edu>.

*Proceedings of the $43^{rd}$ International Conference on Machine Learning*, Seoul, South Korea. PMLR 306, 2026. Copyright 2026 by the author(s).

Figure 1 illustrates the scale of the problem. The number of accepted papers at major AI conferences has grown rapidly. Throughout this expansion, a substantial proportion of papers, frequently exceeding 30% at major venues, explicitly claim state-of-the-art performance in their abstracts.

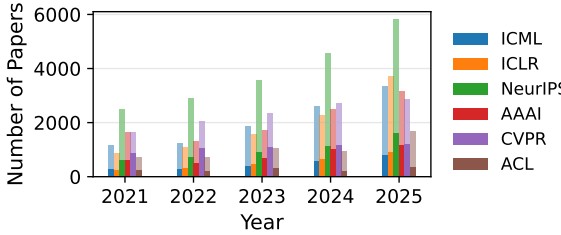

*Figure 1.* Published papers mentioning "state-of-the-art" in abstracts across major AI conferences (2021–2025). Darker bars indicate SOTA mentions, and lighter bars indicate other papers. In this study, the empirical analysis in Section 4 focuses on benchmarks that aggregate across multiple tasks or datasets.

Please refer to Appendix A for details on the terminology, scope, and methodology used in Figure 1.

A mean score across $k$ tasks establishes only that a model ranks first on average. It does not show that the model wins on most tasks or would retain its rank under different dataset compositions. Demšar (2006) documented that statistical tests are routinely misapplied or omitted entirely. Prior work demonstrates that variance arising from data sampling, hyperparameter selection, and test set construction is often comparable to reported improvements (Bouthillier et al., 2021). However, marginal gains in aggregate scores continue to be interpreted as evidence of robust superiority.

This *claim-evidence gap* is pervasive across modalities. In natural language processing, rankings are demonstrably sensitive to task composition and prompt formatting, often reversing leadership based on minor perturbations (Dehghani et al., 2021; Alzahrani et al., 2024; Mirzadeh et al., 2025; Gema et al., 2025). Computer vision benchmarks exhibit similar volatility, where top-ranked models frequently degrade under distribution shifts (Recht et al., 2019; Ozbulak et al., 2024). Analogous patterns plague time series forecasting, where selective dataset reporting allows a vast majority of methods to claim spurious superiority (Roque et al., 2025). Collectively, these findings confirm that benchmark-based rankings are inherently fragile.

Despite widespread awareness, claim language remains unchanged. Researchers continue to claim "SOTA" based on the same thin evidence that prior work has shown to be unreliable. Thus, this paper makes three contributions:

- *We identify a claim-evidence gap.* A mean score establishes only that a model ranks first on average. It does not establish that the model wins consistently, improves meaningfully, or would retain its rank under different conditions. However, claiming "state-of-the-art" carries implicit assumptions beyond mean score superiority. Our diagnostics make these assumptions explicit and testable.

- *We show the gap is pervasive.* We apply elementary statistical diagnostics to ten public leaderboards. We find that more than half of top-model comparisons lack support for at least one commonly assumed property of superiority. This pattern holds across benchmarks that aggregate semantically distinct tasks, separate datasets within a single task, or different evaluation dimensions.

- *We argue that closing the gap requires institutional norms.* The diagnostics we apply require no new experiments. The persistence of the problem despite widespread awareness confirms that the barrier is institutional, not technical. Individual authors cannot shift equilibria alone, but venue-level guidelines can.

Our analysis addresses benchmark-based SOTA claims, that is, comparisons where models are ranked by aggregate scores across tasks. This structure dominates major venues, though some work uses application-specific evaluation. We are not criticizing benchmark design. Public benchmarks have accelerated progress by enabling reproducible comparison. Our concern is with interpretation: **claim language that exceeds what the evidence supports.**

We argue that claim language in the ML literature often exceeds what the computed evidence supports. Papers compute factual evidence but use language reserved for robust superiority. We support this argument by analyzing ten major benchmarks drawn from public leaderboards. The diagnostic methods apply to any benchmark comparison, whether published on a leaderboard or reported in a paper table. **We call for venue-level norms that encourage claim-evidence alignment.** This requires no new experiments, only honest interpretation of existing results.

**Conflict of Interest Disclosure.** No financial conflicts.

## 2. Related Work

### 2.1. Statistical Comparison Methods

Prior work has established statistical frameworks for model comparison centered on the Wilcoxon signed-rank test (Wilcoxon, 1945; Fix & Hodges, 1955; Woolson, 2008), which remains a standard recommendation for pairwise evaluation. Demšar (2006) showed that paired t-tests over cross-validation folds inflate Type I error, leading to a preference for non-parametric alternatives. In practice, critical-difference diagrams (Demšar, 2006) are used in time-series studies (Ismail Fawaz et al., 2019; Oh et al., 2025).

This analysis was extended to post-hoc comparisons and multiple testing corrections (García & Herrera, 2008; Carterette, 2017; Dror et al., 2018). However, we address a distinct problem: even when appropriate tests are applied, claim language often exceeds what the evidence supports.

Variance from hyperparameter selection often rivals reported improvements (Bouthillier et al., 2021). Similarly, cross-validation error estimation exhibits structural variance dominated by training set sensitivity (Rodriguez et al., 2010). Also, Berrar (2024) showed that marginal $p$-values yield low replication probability. Nevertheless, adoption remains limited. Our approach aligns with this diagnostic perspective without necessitating experimental redesign. Instead, it focuses on the implementation of disciplined reporting.

### 2.2. Benchmark Fragility

Dehghani et al. (2021) introduced the "benchmark lottery" to describe how rankings depend on task selection. Roque et al. (2025) quantified dataset selection bias in time series forecasting. Minor protocol changes such as prompt formatting can cause substantial ranking shifts (Alzahrani et al., 2024; Sclar et al., 2024). Human-preference evaluation shows similar sensitivity. Position and verbosity biases change rankings when not controlled (Zheng et al., 2023). Furthermore, Mirzadeh et al. (2025) showed that performance varies substantially across semantically identical questions. Gema et al. (2025) found systematic errors in MMLU that change model rankings when corrected. Similarly, Recht et al. (2019) documented accuracy drops under distribution shift even on mature benchmarks.

Rigorous reporting practices exist, though they remain rare. Chatbot Arena (Chiang et al., 2024) reports Bradley-Terry coefficients with confidence intervals rather than point rankings. FEV-Bench (Shchur et al., 2025) provides bootstrap confidence intervals on win rates, revealing that apparent gaps between top models are often not statistically distinguishable. These examples demonstrate that uncertainty quantification is feasible at negligible cost. Their rarity reflects institutional barriers rather than technical ones.

While prior research has diagnosed benchmark fragility and proposed technical remedies (e.g., new metrics or protocols), the language used to describe benchmark results has received far less scrutiny. We argue that SOTA designations must reflect the statistical strength of available evidence.

# 3. Matching Claims to Evidence

*"Extraordinary claims require extraordinary evidence."*

– Carl Sagan (1979)

Claims of greater scope require proportionally stronger evidence (Sagan, 1979). In ML, however, broad claims of model superiority often rely on single point estimates. We argue that the strength of a claim should be commensurate with the strength of the evidence.

## 3.1. Problem Formulation

Let $\mathcal{D} = \{d_1, \ldots, d_k\}$ denote a benchmark comprising $k$ evaluation units, called tasks, subjects, datasets, or dimensions depending on the domain (see Appendix A). Let $\mathcal{M}$ be a set of $M$ models. For a model $m \in \mathcal{M}$ on dataset $d$, let $s_m(d) \in \mathbb{R}$ denote the loss or error rate, assuming lower is better, without loss of generality.

Under the conventional criterion, model $A$ is declared superior to $B$ if and only if:

$$\bar{s}_A < \bar{s}_B \quad \text{where} \quad \bar{s}_m = \frac{1}{k} \sum_{d \in \mathcal{D}} s_m(d) \qquad (1)$$

This inequality is the sole basis for most rankings. The question is whether a lower mean score reflects genuine superiority or statistical artifact.

In this paper, we argue that the per-dataset view reveals patterns that aggregate scores obscure. For any model pair $(A, B)$, we define the per-dataset performance differential as:

$$\delta_d(A, B) = s_B(d) - s_A(d) \qquad (2)$$

where $\delta_d > 0$ indicates that model $A$ outperforms model $B$ on dataset $d$. Given $M$ models on a benchmark, we form all $\binom{M}{2} = \frac{M(M-1)}{2}$ pairwise comparisons. For each pair, we designate the model with the lower (or higher) mean score as the winner and ask, "Does the evidence really support claims beyond this factual observation?"

## 3.2. Deconstructing SOTA Claims

When a model is described as "state-of-the-art," readers may expect one or more of the following properties to hold:

- The model achieves a better average score.

- The improvement is meaningfully large.

- The model wins consistently across tasks.

- The ranking is stable under minor perturbations.

Current practice verifies only the first. The remaining three are assumed but often ignored. These properties are independent: a model pair may satisfy some while failing others.

Our diagnostics make these implicit assumptions explicit and testable, rather than prescribing requirements:

1. Is the performance gap meaningful, or is it indistinguishable from noise?

2. Does the top model actually win on most tasks, or does it lose more often than it wins?

3. Is the ranking stable, or does removing one dataset reverse the conclusion?

We do not argue that every criterion must be satisfied. We argue that when any of these elementary tests fails, the comparison is fragile, and the strength of the claim should be tempered accordingly. A model that ranks first by mean score but loses on most tasks has demonstrated only a narrow form of superiority. It is the best on average, under one particular aggregation, on one particular set of datasets. Claiming broader superiority requires broader evidence.

## 3.3. Three Perspectives on Fragility

We do not propose novel statistical methods. Instead, we apply three well-established concepts from statistics to test whether observed rankings hold under minimal scrutiny.

**Dimension 1: Magnitude (Effect Size).** The implicit claim that $A$ shows meaningful improvement can be tested via effect size. Standardized effect sizes address the limitation of statistical significance alone, which conflates effect magnitude with sample size (Cohen, 1988). Consequently, effect size reporting has been recommended for machine learning evaluation (Demšar, 2006; Dror et al., 2018):

$$d(A, B) = \frac{\bar{\delta}}{s_\delta} \quad \text{where} \quad \bar{\delta} = \frac{1}{k} \sum_d \delta_d, \quad s_\delta = \text{std}(\{\delta_d\}) \tag{3}$$

Cohen (1988) established conventions for interpreting $d$, characterizing $0.2$ as small, $0.5$ as medium, and $0.8$ as large. Cohen's $d$ compares the performance gap between two models to the variation of their scores across tasks. A value of $d = 0.2$ means the gap only slightly exceeds task-to-task variation. The default $\tau_d = 0.2$ therefore sets a small effect threshold. Failure indicates that the gap is negligible relative to cross-task variance. Two models may differ in mean yet remain practically indistinguishable.

**Dimension 2: Consistency (Win Rate).** The assumption that $A$ outperforms $B$ on most tasks can be tested via the win rate. This approach has precedent in machine learning as pairwise comparison (Demšar, 2006) and in clinical trials as the win ratio (Pocock et al., 2012; Redfors et al., 2020):

$$\text{WR}(A, B) = \frac{1}{k} \sum_{d \in \mathcal{D}} \mathbb{1}[\delta_d(A, B) > 0] \qquad (4)$$

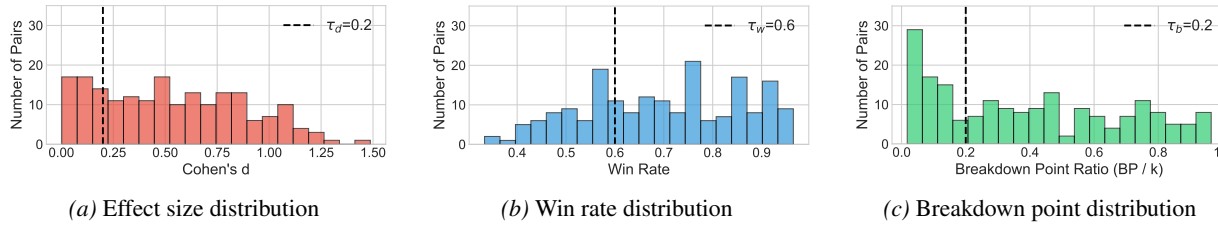

*(a)* Effect size distribution        *(b)* Win rate distribution        *(c)* Breakdown point distribution

*Figure 2.* Distribution of diagnostic metrics across all pairwise comparisons on HELM MMLU.

Bouthillier et al. (2021) recommend a threshold of $P(A > B) \geq 0.75$ for claiming superiority, whereas our default $\tau_w = 0.6$ is deliberately more lenient. The win rate measures how often the supposedly superior model wins across tasks. Failure indicates that $A$ outperforms on only a minority of tasks despite a higher average score.

**Dimension 3: Stability (Breakdown Point).** The implicit claim that $A$'s superiority is robust can be tested via the breakdown point. This concept from robust statistics measures the fraction of observations that must be removed to change a conclusion (Hampel, 1971; Rousseeuw & Leroy, 1987). Influence diagnostics based on leave-one-out analysis have been applied in meta-analysis to assess whether conclusions depend on individual studies (Viechtbauer & Cheung, 2010):

$$\text{BP}(A, B) = \frac{1}{k} \min \left\{ |S| : S \subset \mathcal{D},\ \bar{s}_A^{(-S)} \geq \bar{s}_B^{(-S)} \right\} \quad (5)$$

where $\bar{s}^{(-S)}$ is the mean score excluding subset $S$.

In robust statistics, estimators with breakdown points below $0.2$ are considered fragile (Rousseeuw & Leroy, 1987). Our default $\tau_b = 0.2$ adopts this convention. The breakdown point measures how many datasets must be removed before the ranking reverses. Failure indicates that the ranking depends on a small subset of datasets, so removing a few favorable results can reverse the conclusion.

### 3.4. Fragility Rate

To quantify how often these assumptions lack support, we define the fragility rate $\mathcal{F}$ as the proportion of model pairs where the mean-score winner fails at least one elementary test. For all pairs $(A, B)$ where $\bar{s}_A < \bar{s}_B$:

$$\mathcal{F} = \frac{1}{n} \sum_{(A,B)} \mathbb{1} \left[ \text{WR} \leq \tau_w\ \vee\ d \leq \tau_d\ \vee\ \text{BP} \leq \tau_b \right] \quad (6)$$

We do not propose these diagnostics as new criteria that models must satisfy, nor do we argue that failing any test disqualifies a claim. Instead, we use them to reveal how often current claims rest on fragile evidence. The fragility rate quantifies the problem rather than prescribing a solution.

## 4. Empirical Findings

### 4.1. Case Analysis: HELM MMLU

We begin with Massive Multitask Language Understanding (MMLU), a 57-subject benchmark for language model evaluation (Hendrycks et al., 2021). Here, we use the MMLU evaluation provided by the Holistic Evaluation of Language Models (HELM) framework[1] (Liang et al., 2023).

As a static benchmark, MMLU has well-documented limitations, including susceptibility to contamination (Xu et al., 2024; Balloccu et al., 2024) and sensitivity to answer ordering (Alzahrani et al., 2024). Paraphrasing questions causes substantial accuracy drops despite preserving semantic content (Lunardi et al., 2025).

**Diagnostic Distributions.** Figure 2 shows the distribution of each diagnostic metric across all model pairs. The effect size distribution (a) shows wide variation with some pairs exhibiting negligible differences relative to cross-task variance. The win rate distribution (b) reveals that a nontrivial fraction of mean-score winners lose on more than 40% of tasks. The breakdown point distribution (c) is notably left-skewed. Many comparisons depend on a small subset of datasets to maintain their ranking which indicates that conclusions would reverse under minor perturbations to the benchmark composition.

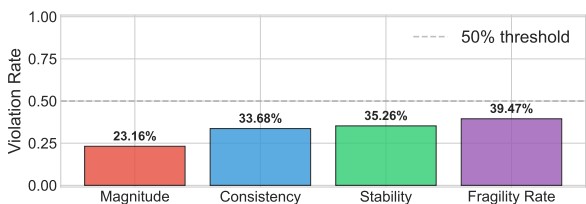

*Figure 3.* Summary of violation rates for each diagnostic test on HELM MMLU. Fragility rate indicates the proportion of model pairs where at least one test fails.

These distributions illustrate that the three tests capture distinct failure modes. For instance, a model pair may show consistent wins but a small effect size. Alternatively, it may exhibit large effects but poor stability. This heterogeneity

---

[1] https://crfm.stanford.edu/helm/mmlu/

underscores why aggregate scores alone cannot characterize the reliability of a comparison.

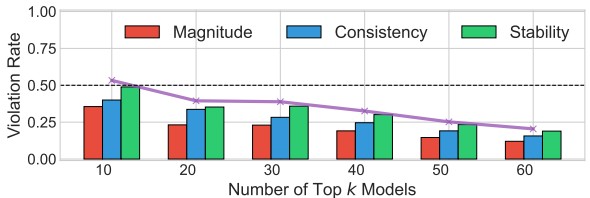

*(a) By number of top-ranked models*

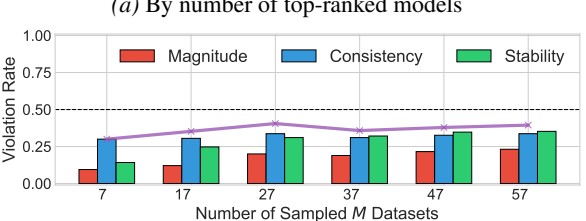

*(b) By number of sampled datasets*

*Figure 4.* Violation rates on HELM MMLU under varying analysis conditions. The purple line indicates the overall fragility rate.

Figure 3 summarizes the violation rates across all pairwise comparisons among the top 20 models. Each test identifies a distinct failure mode: inconsistent wins, negligible effect sizes, or dependence on favorable dataset subsets. The overall fragility rate reflects comparisons where the mean-score winner lacks support on at least one of these dimensions.

**Robustness to Analysis Scope.** Fragility is not uniform across the ranking. Figure 4 (a) demonstrates that violation rates are highest among the top 10 models, precisely where SOTA claims concentrate. As additional models are included, fragility decreases. Expanding the analysis from the top 10 to the top 60 models increases the number of pairwise comparisons from 45 to 1,770. However, the fragility rate declines rather than stabilizes. This pattern suggests that distinctions among top performers are less robust than the distinctions between top-tier and middle-tier models.

A natural hypothesis is that fragility reflects insufficient task coverage. Figure 4 (b) tests this by subsampling datasets. Violation rates remain stable regardless of whether 7 or 57 tasks are included. Adding tasks does not resolve instability when tasks share similar characteristics. Consequently, the observed fragility reflects structural heterogeneity rather than an insufficient sample size.

### 4.2. Are the Tests Too Strict?

A natural concern is that our thresholds are too demanding, artificially inflating fragility rates. We do not argue that all three tests must pass to claim progress. Rather, we measure how often at least one commonly assumed property of superiority fails to hold.

This concern can be addressed by three observations. First, the individual thresholds are lenient. A win rate of $0.6$ requires success on only 60% of tasks, well below the probability of superiority threshold $P(A > B) \geq 0.75$ recommended by Bouthillier et al. (2021) for claiming one algorithm outperforms another. Cohen's $d = 0.2$ corresponds to the minimum threshold for a small effect (Cohen, 1988). A breakdown point ratio of $0.2$ is the standard threshold below which estimators are considered fragile in robust statistics (Hampel, 1971; Rousseeuw & Leroy, 1987).

Second, the tests capture distinct failure modes. Some pairs exhibit consistent wins with small effect sizes, while others demonstrate large effects but poor robustness. This pattern echoes the "benchmark lottery" phenomenon documented by Dehghani et al. (2021) where model rankings depend on which aspect of performance is emphasized. Therefore, mean score winners fail through distinct failure modes rather than a single common mechanism.

Third, fragility rates remain high even when thresholds are relaxed. Figure 5 illustrates this robustness for HELM MMLU. As each threshold varies from lenient to strict, individual violation rates change substantially. However, the overall fragility rate remains above 30% across all single-parameter configurations. Sensitivity analysis for all ten benchmarks (Appendix D) confirms that fragility rates remain above 50% even under the most lenient individual threshold tested. This robustness to threshold choice is consistent with Demšar (2006)'s broader call for explicit statistical analysis in benchmark comparisons.

In summary, the frequency with which at least one property fails reflects not excessive strictness but the fact that mean score superiority does not reliably imply consistent, meaningful, and robust superiority. This mismatch constitutes the claim-evidence gap identified in our analysis.

### 4.3. Cross-Domain Fragility Analysis

To test generality, we conducted a cross-domain analysis across ten diverse benchmarks spanning Large Language Model, Natural Language Processing, Computer Vision, Audio, Tabular, and Time Series domains. These benchmarks aggregate performance over units that range from subject areas to individual datasets, but share the same mathematical structure, as explained in Section 3.

We selected benchmarks that are widely adopted and peer-reviewed. HELM MMLU is a widely adopted benchmark for evaluating LLMs (Hendrycks et al., 2021; Gema et al., 2025). The remaining benchmarks were introduced at top-tier venues (Table 1). Please refer to Appendix B for publication details.

Our analysis builds upon these benchmarks. We do not critique their design. Instead, we examine whether reporting

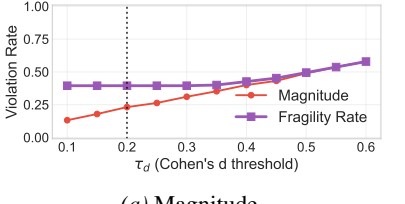 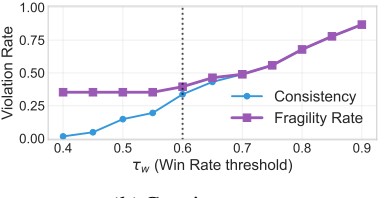 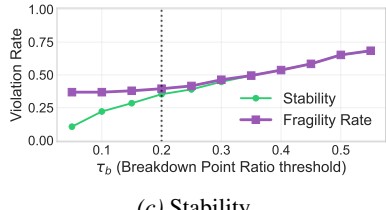

*(a)* Magnitude        *(b)* Consistency        *(c)* Stability

*Figure 5.* Sensitivity to threshold selection on HELM MMLU. Each panel varies one threshold while holding others at defaults.

practices accurately communicate ranking uncertainty. We reconstructed pairwise comparison matrices using public performance logs with a cut-off date of December 31, 2025. Table 1 provides the detailed specifications for each dataset. Refer to Appendix B for details regarding each leaderboard.

For each benchmark, we selected the top 20 models based on mean score (or the full set if fewer than 20 were available). We then computed all pairwise comparisons, yielding up to $\binom{20}{2} = 190$ pairs per benchmark. For each pair where model $A$ outperformed model $B$ on average, we applied the three diagnostics defined in Section 3.

For each pair, the model with the better mean score is designated as the winner. We then compute three diagnostics per pair: Cohen's $d$, win rate, and breakdown point ratio. The reported mean and standard deviation summarize the distribution of these values across all pairs. We employed deliberately lenient thresholds ($\tau_w = 0.6$, $\tau_d = 0.2$, $\tau_b = 0.2$). We do not demand large, definitive differences, but rather test for the presence of any reliable difference. Appendix C and Appendix D detail each benchmark and examine the robustness of our results to threshold choices.

Table 1 summarizes our analysis across ten leaderboard examples. Across more than 1,000 pairwise comparisons spanning six domains and ten benchmarks, a substantial fraction of pairwise comparisons exhibit at least one violation. The diagnosed fragility appears in most comparisons.

**The bottleneck varies.** In language model benchmarks such as HELM MMLU and LiveBench, rankings tend to be stable, likely reflecting architectural similarity among top models. Tabular benchmarks exhibit higher fragility because gradient boosting and neural approaches each dominate different dataset types, and no single method consistently outperforms across the heterogeneous problems in TabArena. Time series forecasting shows the highest fragility. This domain aggregates datasets from totally different generative processes. Therefore, rankings frequently reverse depending on the specific subset of domains prioritized.

**Metric and dataset choices amplify fragility.** TSFM-Bench evaluates identical models on identical datasets under both MAE and MSE. The MSE variant shows substantially

higher fragility because squared errors amplify outlier sensitivity. This allows a single anomalous dataset to dominate aggregate rankings. Similarly, Open ASR and TabArena Multiclass each use only eight datasets. This limits statistical power and makes rankings sensitive to the inclusion or exclusion of any single evaluation set.

**Score gaps do not determine fragility.** A model that leads by a narrow margin but wins consistently across tasks exhibits low fragility. Conversely, a model with a larger aggregate lead driven by dominance on a few outlier tasks can see its ranking collapse when those tasks are excluded. This distinction is invisible when only aggregate scores are reported. Thus, fragility arises from the distribution of performance rather than aggregate proximity.

### 4.4. When Do These Concerns Apply?

The claim-evidence gap is not an artifact of any single benchmark but a structural feature of how leaderboards aggregate heterogeneous signals into scalar rankings. Any research reporting aggregate performance across multiple datasets or tasks faces the same fundamental question regarding whether the reported evidence supports the stated claim. We emphasize a narrow but important point regarding scope. Not every paper requires comprehensive multi-task analysis. A method evaluated on only two or three datasets naturally supports a limited claim, whereas assertions of state-of-the-art performance imply broader evidence. Problems arise not from small evaluations themselves but from pairing narrow evidence with expansive claim language.

The diagnostics we discuss, scale naturally with evaluation size and incur negligible cost. Even in small comparisons, reporting how often one method wins or where improvements fail provides meaningful context. Our concern therefore lies not with evaluation rigor but with claim calibration. Describing results without a claim-evidence gap is appropriate when supported by point estimates alone. Describing the same evidence as a "robust state-of-the-art improvement" creates the gap we aim to make visible. Authors should assess whether their reporting language exceeds their evidence, regardless of whether the work involves a public leaderboard or standard benchmark.

*Table 1.* Cross-domain fragility analysis using ten public leaderboards (Cut-off: Dec 31, 2025). Values in parentheses are standard deviations across model pairs. Metrics and scales vary across benchmarks. Therefore, point estimate gaps are not directly comparable.

| | HELM MMLU (Hendrycks et al., 2021) | LiveBench (White et al., 2025) | Open ASR (Srivastav et al., 2025) | Open VLM (Duan et al., 2024) | VBench (Huang et al., 2024) | TabArena _Binary (Erickson et al., 2025) | TabArena _Multiclass (Erickson et al., 2025) | TabArena _Regression (Erickson et al., 2025) | TSFM (MAE) (Li et al., 2025) | TSFM (MSE) (Li et al., 2025) |
|---|---|---|---|---|---|---|---|---|---|---|
| Domain | NLP/LLM | LLM | Speech | Vision-Language | Video Gen. | Tabular | Tabular | Tabular | Time Series | Time Series |
| Snapshot | 2025/01/10 | 2025/12/23 | 2025/10/30 | 2025/09/17 | 2025/08/15 | 2025/12/11 | 2025/12/11 | 2025/12/11 | 2025/08/03 | 2025/08/03 |
| Number of Models (Total) | 20 (79) | 20 (58) | 20 (39) | 20 (284) | 20 (68) | 20 (47) | 20 (47) | 20 (47) | 14 (17) | 14 (17) |
| Number of Datasets (Total) | 57 (57) | 21 (21) | 8 (8) | 29 (31) | 16 (16) | 30 (30) | 8 (8) | 13 (13) | 72 (84) | 72 (84) |
| Metric | Match ($\uparrow$) | Accuracy ($\uparrow$) | WER ($\downarrow$) | Accuracy ($\uparrow$) | Accuracy ($\uparrow$) | 1-AUC ($\downarrow$) | Log Loss ($\downarrow$) | RMSE ($\downarrow$) | MAE ($\downarrow$) | MSE ($\downarrow$) |
| **Point Estimate** | | | | | | | | | | |
| Score Gap (1st-2nd) | 0.0007 | 1.0424 | 0.1075 | 1.9172 | 0.0113 | 0.0001 | 0.0012 | 91.2076 | 0.0031 | 0.0077 |
| Score Gap (1st-2nd, %) | 0.08% | 1.41% | 1.87% | 1.05% | 1.45% | 0.09% | 0.46% | 1.41% | 0.74% | 1.32% |
| **Statistical** | | | | | | | | | | |
| Number of Pairs | 190 | 190 | 190 | 190 | 190 | 190 | 190 | 190 | 91 | 91 |
| Wilcoxon $p < 0.05$ (%) | 74.74% | 49.47% | 8.95% | 64.21% | 35.79% | 43.68% | 7.37% | 38.42% | 58.24% | 64.84% |
| **Magnitude** | | | | | | | | | | |
| Cohen's d | 0.53 (0.35) | 0.51 (0.34) | 0.40 (0.27) | 0.28 (0.12) | 0.40 (0.22) | 0.33 (0.21) | 0.32 (0.30) | 0.28 (0.04) | 0.33 (0.20) | 0.23 (0.16) |
| – Violation Rate (%) | 23.16% | 21.58% | 30.00% | 25.79% | 21.58% | 27.89% | 43.68% | 3.16% | 28.57% | 54.95% |
| **Consistency** | | | | | | | | | | |
| Win Rate (%) | 69.88 (16.10) | 63.86 (13.47) | 64.41 (16.48) | 74.81 (17.40) | 63.22 (14.33) | 65.65 (13.26) | 60.20 (18.73) | 60.08 (24.32) | 63.84 (14.35) | 54.01 (20.52) |
| – Violation Rate (%) | 33.68% | 39.47% | 34.74% | 20.53% | 34.21% | 37.89% | 41.58% | 46.84% | 46.15% | 60.44% |
| **Stability** | | | | | | | | | | |
| BreakPoint rate (%) | 39.45 (29.24) | 34.66 (23.70) | 33.68 (22.86) | 37.42 (30.60) | 28.52 (19.78) | 21.32 (17.00) | 27.57 (21.76) | 29.64 (28.13) | 22.41 (19.82) | 17.05 (21.87) |
| – Violation Rate (%) | 35.26% | 35.79% | 32.11% | 40.53% | 47.37% | 58.95% | 51.58% | 54.74% | 57.14% | 74.73% |
| **Fragility Rate (%)** | 39.47% | 44.21% | 45.79% | 42.11% | 53.68% | 61.05% | 58.95% | 58.42% | 61.54% | 74.73% |

## 5. Alternative Views

### 5.1. Mean Scores Suffice for Practical Decisions

A natural objection is that practitioners care only about which model performs best on average. This concern reflects legitimate practical constraints. Engineers facing deployment deadlines cannot run extensive statistical analyses for every model comparison. For many applications, selecting the top-ranked model by average performance is reasonable. We do not dispute that mean scores provide useful signal.

However, this objection conflates two distinct claims. The first is that mean scores help identify good models. The second is that mean scores justify superiority claims. We accept the first but reject the second. A model that ranks first by average accuracy can still fail on tasks that matter in deployment. Performance on MMLU varies across its 57 subjects (Hendrycks et al., 2021). Small changes in question format can shift rankings by several positions (Alzahrani et al., 2024). For instance, a practitioner selecting a model for chemistry tutoring based on aggregate MMLU scores may encounter weak accuracy on chemistry questions despite strong overall performance. Roque et al. (2025) demonstrate that dataset selection alone can make a large fraction of methods appear "best in class." If practitioners need only average performance, authors should say so explicitly.

The phrase "SOTA" implies broad superiority that mean scores alone do not establish. When benchmark tasks are heterogeneous, a low win rate shows that the aggregate winner specializes in a subset of tasks, which the mean obscures. This supports transparent reporting. Whether diagnostic violations reflect benchmark heterogeneity or weak evidence is a question of benchmark design that we do not address here (see Appendix B).

### 5.2. More Tasks Solve the Problem

A common counterargument suggests that fragility arises from insufficient task coverage. Proponents of this view argue that if rankings are unstable, the remedy is simply to increase the number of tasks. Averaging over a larger set of independent measurements decreases variance. This principle is statistically sound. If benchmark tasks provided independent evidence regarding model capability, adding more tasks would indeed stabilize rankings.

However, this premise fails to hold in practice. Benchmark tasks are rarely independent. Instead, tasks within the same benchmark frequently share design choices, data sources, or evaluation protocols that induce high correlation (Dehghani et al., 2021). Different subsets of tasks produce different winners. If tasks were truly independent, the aggregate rank-

ing should remain stable regardless of the specific subset examined. This sensitivity to task selection shows that task count alone does not guarantee stability (Wang et al., 2024; Mirzadeh et al., 2025; Gema et al., 2025).

The critical factor is whether tasks provide genuinely independent evidence. Even superficial changes to benchmark questions, such as paraphrasing or answer reordering, can cause substantial performance drops despite preserving semantic content (Lunardi et al., 2025; Alzahrani et al., 2024). This finding suggests that aggregated scores often reflect sensitivity to surface features rather than purely underlying capability. Therefore, adding more tasks with similar characteristics does not address the fundamental issue. The solution lies not in creating larger benchmarks, but in accurately reporting the limitations of the results.

### 5.3. The Community Already Knows This

Another view accepts our diagnosis but questions its novelty. Experienced researchers understand that benchmark rankings are noisy and that small differences may not reflect real performance gaps. If awareness is widespread, what does this paper contribute?

The objection has partial merit. Demšar (2006) identified the need for proper statistical comparison nearly two decades ago. Dror et al. (2019) showed that standard methods fail to reach decisions in half of Deep Neural Network (DNN) comparisons in NLP. Bouthillier et al. (2021) proposed probabilistic comparisons. The methodology exists. The problems are known.

However, knowledge has not produced change. Lipton & Steinhardt (2019) identified concerns in ML scholarship, and misuse of language such as unsupported SOTA claims remains common. Our contribution is to quantify this gap and provide a basis for institutional action. Three primary factors reinforce the status quo. First, publication pressure rewards bold claims. Consequently, authors who offer nuanced or hedged conclusions face a competitive disadvantage. Second, subconscious biases in interpretation persist. Even factual phrases like "ranks first on this benchmark" may be read as implicit superiority claims because the community lacks standardized terms to distinguish narrow from broad evidence. Finally, enforcement mechanisms are largely absent. Currently, research venues do not penalize claims that exceed the strength of the provided evidence.

While certain benchmarks have already begun to incorporate uncertainty metrics such as confidence intervals and bootstrap win rates (Chiang et al., 2024; Shchur et al., 2025), the majority still omit statistical significance (Reuel et al., 2024). At a broader level, Ott et al. (2022) report that many benchmarks saturate quickly yet continue to be used for ranking despite reduced discriminative power.

Since these practices require no additional experiments, their rarity confirms that the barrier is primarily institutional rather than technical.

### 5.4. Institutional Change Requires Coordination

A pragmatic objection is that reviewers routinely request SOTA comparisons irrespective of how claims are framed. Therefore, conforming to existing norms remains a rational strategy. Metrics act as proxies that diverge from true objectives under optimization pressure (Thomas & Uminsky, 2022). This divergence highlights precisely why institutional coordination is essential.

Clinical medicine faced a similar problem decades ago and developed two complementary solutions. The CONSORT (Consolidated Standards of Reporting Trials) statement is a checklist requiring authors to disclose how trials were conducted so that readers can assess whether reported effects are trustworthy (Moher et al., 2001; Altman et al., 2001; Hopewell et al., 2025). Additionally, the GRADE (Grading of Recommendations Assessment, Development and Evaluation) framework separates evidence quality from recommendation strength (Guyatt et al., 2008; Granholm et al., 2019; Schünemann et al., 2023). Both frameworks improved practice not by requiring new experiments but by mandating the honest reporting of existing ones.

Our community has already begun to follow this path. The adoption of the ML Reproducibility Checklist (Pineau et al., 2021) and the REFORMS (Recommendations for Machine-learning-based Science) framework (Kapoor et al., 2024) shows that the community can coordinate around shared reporting norms. However, current standards prioritize replicability over comparative validity. Because unilateral adoption of rigorous ranking criteria incurs a competitive disadvantage, venue-level requirements must expand to cover SOTA claims to break this equilibrium. Our contribution is not the diagnosis. Instead, it is the argument that awareness without enforcement changes nothing.

## 6. Limitations

- **Model Selection:** We analyze only top-ranked models. SOTA claims concentrate among top models, making this the relevant population. However, rankings further down may exhibit different fragility patterns.

- **Threshold Choice:** We set thresholds for three tests at lenient levels grounded in prior literature. Our sensitivity analysis shows that conclusions hold across reasonable variations. However, we do not claim these thresholds are uniquely correct.

- **Single-Metric Focus:** Our analysis examines benchmarks where a single aggregate metric determines rank-

ings. Evaluation frameworks reporting multiple metrics simultaneously may exhibit different patterns.

- **Temporal Scope:** Our analysis reflects a single snapshot of each benchmark. Public leaderboards evolve as models are submitted and deprecated. We do not track how fragility changes over time.

- **Causal Explanation:** We document fragility but do not explain its sources. Why some benchmarks exhibit higher fragility likely depends on task diversity, metric choice, and model similarity.

# 7. Conclusion and Recommendations

Machine learning research has normalized a gap between evidence and claims. Our cross-domain analysis demonstrates that this gap is pervasive. The majority of top-model comparisons lack the statistical support that "SOTA" implicitly promises. The tools to close this gap already exist. Therefore, the barrier is institutional rather than methodological.

## 7.1. For Authors

Claiming "state-of-the-art" carries assumptions that go beyond ranking first by mean score. However, our analysis shows that these assumptions often lack support. Many mean-score winners do not win consistently, do not improve meaningfully, or do not retain their rank under minor perturbations, as shown in Breakdown Point analysis.

We do not propose that authors must pass specific statistical tests. Instead, we propose that claim language should reflect what the evidence actually supports. When aggregate gains are narrow or inconsistent, phrases like "achieves lowest average error" or "ranks first on this benchmark" are more accurate than "state-of-the-art," which serves as a marketing term rather than a precise scientific claim. However, terminology revision alone is insufficient if "ranks first" inherits the same subconscious connotations. Accompanying such claims with win rates and effect sizes prevents any single label from serving as an uncritical proxy for superiority.

## 7.2. For Reviewers

The gap we document can be identified without new experiments. Reviewers need not apply our specific diagnostics. Instead, a simpler question suffices: does the claim language accurately reflect the reported statistics? For instance, a paper describing a method that underperforms on a substantial fraction of tasks should not claim to "significantly outperform" competitors. Verifying whether the interpretation aligns with the evidence is sufficient.

We also encourage reviewers to reconsider standard requests such as "compare with the latest SOTA" or "add more baselines." Such demands can inadvertently pressure authors to pursue fragile leaderboard positions rather than articulating genuine scientific contributions. The critical question is not "does this method surpass all predecessors?" but rather "what is the specific contribution, and is the evidence sufficient to support the claims made?"

## 7.3. For Venues

Public leaderboards significantly influence community norms. Currently, most platforms rank models exclusively by mean scores, thereby implicitly endorsing comparisons based on single point estimates. Leaderboards that incorporate win rates or confidence intervals alongside aggregate scores would render the fragility of close rankings immediately apparent. This transparency would enable users to calibrate their interpretations without imposing additional burdens on authors. Some benchmarks already report uncertainty through confidence intervals or bootstrap win rates, revealing that apparent gaps are often not statistically distinguishable. These practices require no additional burden.

Broader adoption requires institutional support. Reviewer guidelines can ask whether claim language matches the reported evidence. Furthermore, submission checklists should prompt authors to characterize the robustness of their conclusions. These interventions require no new experiments. Instead, they require only honest interpretation.

## 7.4. Concluding Remarks

**Matching Claims to Evidence.** Our goal is to align claim language with the strength of the supporting evidence. In current practice, aggregate rankings are often interpreted as broad superiority, even when the underlying evidence may be narrow or heterogeneous. Reporting language should reflect these limitations, especially when performance varies across tasks or depends on specific evaluation choices.

**Scope and Future Work.** Our analysis focuses on single-metric, multitask benchmarks at a fixed snapshot in time. We do not address single-task settings or multi-metric decision scenarios, where different notions of superiority may apply. Future work includes extending these diagnostics to multi-metric evaluation, tracking fragility over time, and analyzing domain-specific patterns that lead to instability, as well as studying how reviewer–author interactions shape SOTA claims in practice.

**Institutional Context.** The patterns we observe arise from widely adopted evaluation practices rather than isolated cases. Because these practices are shared across the community, changes in reporting are unlikely to occur at the level of individual papers alone. Aligning claims with evidence shifts attention away from marginal ranking differences and toward more stable comparisons.

# Acknowledgements

We thank the maintainers of the public leaderboards and the broader open-source community. Their dedication to democratizing access to evaluation results has not only driven the field forward but also provided the necessary transparency for the critical examination presented in this work.

We are grateful to Prof. Alex Bui and the UCLA Medical Imaging Informatics (MII) group for their support. We also thank the anonymous reviewers for their constructive feedback and discussion, which improved this work.

Dr. YongKyung Oh was supported by the Basic Science Research Program through the National Research Foundation of Korea (NRF) funded by the Ministry of Education (RS-2024-00407852).

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

# A. SOTA Mention Analysis

We analyzed abstracts from six major AI conferences (2021–2025) using metadata from the Paper Copilot repository[2], accessed in early 2026. Papers were filtered to accepted submissions only, excluding rejected, withdrawn, and desk-rejected papers. For AAAI, we included main and special tracks, while for ACL, we excluded Findings papers. It is important to note that our filtered counts may differ slightly from official statistics due to data update timing and track inclusion criteria. Our focus is on cross-conference trends rather than absolute counts.

*Table 2.* Number of accepted papers by conference and year (counts may differ).

| | 2021 | 2022 | 2023 | 2024 | 2025 |
|---|---|---|---|---|---|
| ICML | 1183 | 1233 | 1865 | 2610 | 3342 |
| ICLR | 860 | 1095 | 1575 | 2260 | 3704 |
| NeurIPS | 2508 | 2901 | 3584 | 4562 | 5812 |
| AAAI | 1654 | 1319 | 1720 | 2501 | 3182 |
| CVPR | 1661 | 2062 | 2359 | 2716 | 2871 |
| ACL | 710 | 700 | 1075 | 940 | 1699 |

*Table 3.* Number of papers mentioning SOTA in abstract (based on filtered counts).

| | 2021 | 2022 | 2023 | 2024 | 2025 |
|---|---|---|---|---|---|
| ICML | 264 | 262 | 386 | 566 | 783 |
| ICLR | 246 | 318 | 476 | 629 | 903 |
| NeurIPS | 627 | 716 | 895 | 1143 | 1611 |
| AAAI | 609 | 515 | 666 | 1000 | 1158 |
| CVPR | 885 | 1068 | 1090 | 1147 | 1184 |
| ACL | 232 | 211 | 299 | 193 | 335 |

*Table 4.* Percentage of papers mentioning SOTA in abstract (based on filtered counts).

| | 2021 | 2022 | 2023 | 2024 | 2025 |
|---|---|---|---|---|---|
| ICML | 22.32 | 21.25 | 20.70 | 21.69 | 23.43 |
| ICLR | 28.60 | 29.04 | 30.22 | 27.83 | 24.38 |
| NeurIPS | 25.00 | 24.68 | 24.97 | 25.05 | 27.72 |
| AAAI | 36.82 | 39.04 | 38.72 | 39.98 | 36.39 |
| CVPR | 53.28 | 51.79 | 46.21 | 42.23 | 41.24 |
| ACL | 32.68 | 30.14 | 27.81 | 20.53 | 19.72 |

**Terminology.** Benchmarks across domains use varied terminology for their evaluation units. Natural language processing benchmarks refer to subjects or task categories, while speech recognition benchmarks are organized by distinct corpora. Computer vision benchmarks define evaluation dimensions. Tabular benchmarks involve multiple tasks and datasets, and time series forecasting benchmarks aggregate over individual datasets.

Our framework treats these uniformly as evaluation units $d \in \mathcal{D}$ over which aggregate scores are computed. The mathematical structure, ranking models by mean score across $k$ units, is identical regardless of the unit definition. We use "dataset" and "task" interchangeably as generic terms.

**Scope and Methodology.** We identified SOTA claims by matching the patterns: "state-of-the-art" (with hyphen variants), "state of the art", "SOTA", "SotA" (case-insensitive), and so on. This count includes papers across all evaluation settings, including single-task and single-dataset comparisons. We retain them because the purpose of this section is to characterize the prevalence of SOTA claim language across venues, not to filter by evaluation structure.

Tables 2, 3, and 4 summarize our analysis. However, our goal is to identify cross-conference trends rather than report exact counts. The results show that a substantial fraction of accepted papers explicitly claim state-of-the-art performance in their abstracts, ranging from 20% to over 50% depending on the venue. Our keyword-based approach may underestimate the true rate, as semantically equivalent claims (e.g., "our model achieves the best performance") are not captured.

# B. Benchmark Descriptions

This appendix provides detailed descriptions of the ten benchmarks analyzed in this study. We selected benchmarks that span diverse domains and remain actively maintained as of 2025. This selection shows that the claim-evidence gap appears across the machine learning landscape. With the exception of MMLU, all benchmarks were published in 2024 or 2025. The analysis does not question benchmark design. These benchmarks represent substantial community contributions. The focus is on how downstream users interpret results and make claims.

### B.1. HELM MMLU

The Massive Multitask Language Understanding (MMLU) benchmark (Hendrycks et al., 2021) is a multiple-choice question answering test that covers 57 subjects, including elementary mathematics, US history, computer science, and law. The original paper found that model performance varies substantially across domains and that models are poorly calibrated, often failing to recognize when they are incorrect. Originally published at ICLR 2021, MMLU has become one of the most widely cited evaluation resources in NLP and LLM, with the dataset and evaluation code released under an MIT license.

We use the MMLU evaluation provided by the Holistic Evaluation of Language Models (HELM) framework (Liang et al., 2023). HELM MMLU was introduced to address inconsistent evaluation practices across model developers, providing standardized prompts and full transparency of all raw prompts and predictions.

---

[2]https://github.com/papercopilot/paperlists

The evaluation uses the Multiple Choice Joint adaptation method, where the model directly generates the answer choice as text. The primary metric is Exact Match (EM). Based on the latest update, we collected 79 models across 57 subjects. The leaderboard is available at `https://crfm.stanford.edu/helm/mmlu/`.

## B.2. LiveBench

LiveBench (White et al., 2025) is an LLM benchmark designed to address test set contamination and LLM judge bias. The benchmark contains questions based on recently released math competitions, arXiv papers, news articles, and datasets, with questions added and updated on a monthly basis. Each question has verifiable, objective ground-truth answers, allowing automatic scoring without the use of an LLM judge. The paper was accepted as a Spotlight at ICLR 2025. The code and benchmark data are publicly available under the Apache 2.0 license.

LiveBench originally comprises six categories: mathematics, coding, reasoning, data analysis, instruction following, and language comprehension. To prevent contamination, approximately one-sixth of the questions are replaced in each monthly update, so that the benchmark is fully refreshed roughly every six months. In our analysis, we have included recent updates to the leaderboard and collected 58 models across 21 datasets spanning 7 categories. The leaderboard is available at `https://livebench.ai/`.

## B.3. Open ASR Leaderboard

The Open ASR Leaderboard (Srivastav et al., 2025) is a benchmark for automatic speech recognition systems designed to address the saturation of ASR evaluation with short-form English and the lack of efficiency reporting. The leaderboard compares open-source and proprietary systems across multiple datasets, standardizing text normalization and reporting both Word Error Rate (WER) for transcription quality and inverse Real-Time Factor (RTFx) for inference speed.

The benchmark includes three evaluation tracks: English transcription, multilingual recognition (German, French, Italian, Spanish, and Portuguese), and long-form audio. All evaluation code and dataset loaders are open-sourced. For simplicity, we focused on 8 subjects from the English tasks and evaluated 39 models. The leaderboard is available at `https://huggingface.co/spaces/hf-audio/open_asr_leaderboard`.

The Open ASR Leaderboard is released as a recent preprint and is included because it evaluates frontier speech models from major industry labs alongside open-source systems. This setting provides a practically relevant comparison.

## B.4. Open VLM Leaderboard

The Open VLM Leaderboard is built upon VLMEvalKit (Duan et al., 2024), an open-source toolkit designed for evaluating large vision-language models. VLMEvalKit employs generation-based evaluation for all models, reporting results derived from both exact matching and LLM-based answer extraction. The toolkit paper appeared at ACM Multimedia 2024, and the codebase is distributed under the Apache 2.0 license.

The toolkit automates data preparation, distributed inference, prediction post-processing, and metric calculation. Reflecting the efficiency of the framework, the leaderboard contains over 200 models; however, we selected the top 20 for our analysis. Furthermore, we excluded 'VCR' and 'COCO_VAL' due to missing data. The leaderboard is available at `https://huggingface.co/spaces/opencompass/open_vlm_leaderboard`.

## B.5. VBench

VBench (Huang et al., 2024) is a benchmark suite for evaluating video generative models. The benchmark was developed to address the limitation that existing metrics do not fully align with human perceptions of video quality. Specifically, VBench decomposes video generation quality into 16 evaluation dimensions, covering both video quality aspects, and video-condition consistency aspects. For each dimension, the benchmark provides tailored prompts and evaluation methods for automatic objective assessment. The authors validated alignment with human perception through preference annotations for each dimension. VBench received a Highlight at CVPR 2024. The code is available under the Apache 2.0 license.

We treat each of the 16 dimensions as a separate evaluation criterion, enabling fine-grained analysis of how model rankings vary across different aspects of video generation quality. While the leaderboard allows users to upload their own evaluations, we have chosen 68 models certified by the VBench team. The leaderboard is available at `https://huggingface.co/spaces/Vchitect/VBench_Leaderboard`.

### B.6. TabArena

TabArena (Erickson et al., 2025) is a living benchmark for machine learning on tabular data, designed to address the limitations of static benchmarks that fail to update when flaws are discovered or new models are released. The benchmark features 51 manually curated datasets, covering binary classification, multiclass classification, and regression tasks across diverse domains. It was presented as a Spotlight at the NeurIPS 2025 Datasets and Benchmarks Track. The code and precomputed results are released under the Apache 2.0 license.

The benchmark employs nested cross-validation with dataset-specific repetition strategies to guard against randomness. The official leaderboard uses an Elo rating system for aggregation, utilizing task-appropriate metrics: ROC AUC for binary classification, log-loss for multiclass classification, and RMSE for regression. A key finding is that post-hoc ensembling of hyperparameter configurations is essential to evaluate models at their full potential.

For our analysis, we evaluate each task type separately rather than using the aggregated Elo scores, treating ROC-AUC error, log-loss, and RMSE as independent evaluation criteria. We denote TabArena_Binary, TabArena_Multiclass, and TabArena_Regression as binary classification (30), multiclass classification (8), and regression tasks (13), respectively. The leaderboard is available at `https://tabarena.ai/`.

### B.7. TSFM-Bench

TSFM-Bench (Li et al., 2025) is a benchmark for evaluating Time Series Foundation Models (TSFMs) pre-trained on massive heterogeneous time series data. It addresses the issue of incompatible evaluation settings in existing TSFMs, which complicates fair comparison. TSFM-Bench provides standardized experimental protocols for dataset splitting, loading, normalization, and few-shot sampling. The paper was published at KDD 2025.

The benchmark covers 21 multivariate datasets across diverse domains with varied statistical characteristics. The primary metrics are Mean Absolute Error (MAE) and Mean Squared Error (MSE), computed per prediction horizon. For our analysis, we treat MSE and MAE as separate evaluation criteria and exclude model-dataset pairs with missing values caused by model incompatibility with certain prediction settings. Additionally, we consider each prediction horizon as an independent task. Thus, our analysis includes 14 models across 18 datasets (excluding "FRED-MD", "NN5", and "Wiki2000_96"). For further details, please refer to the original paper and the official repository at `https://github.com/decisionintelligence/TSFM-Bench`.

## C. Summary of Individual Results

This section reports violation rates for each diagnostic test across the ten analyzed benchmarks. The code is available at `https://github.com/yongkyung-oh/SOTA`.

For each benchmark, we first rank models by mean score under the conventional criterion and select the top 20 models (or fewer if unavailable). We then form all $\binom{M}{2}$ pairwise comparisons. Within each pair, the model with the higher mean score is designated as the potential winner. Subsequently, we compute three diagnostics from the per-dataset score matrix.

We did not classify individual model entries by publication status. The analysis uses the top 20 models by mean score on each leaderboard without filtering for peer-reviewed publications. Top-ranked positions are dominated by frontier models from well-resourced organizations with publications at major venues.

- **Cohen's $d$.** We compute the mean performance differential across datasets and divide it by the standard deviation. This process yields a scale-free measure of effect magnitude. Values below $\tau_d = 0.2$ indicate that the performance gap is negligible relative to cross-task variability.

- **Win Rate.** For each pair, we calculate the fraction of datasets where the potential winner outperforms the loser. We exclude ties from the numerator. A win rate below $\tau_w = 0.6$ indicates that the advantage in mean score does not translate to consistent per-task superiority.

- **Breakdown Point.** We determine the minimum proportion of datasets whose removal reverses the mean-score ranking. Datasets are sorted by the potential winner's advantage and removed greedily in descending order until the ranking flips. Since the mean is additive, this greedy procedure yields the exact minimum. A breakdown point below $\tau_b = 0.2$ indicates that the ranking relies on a small subset of favorable datasets.

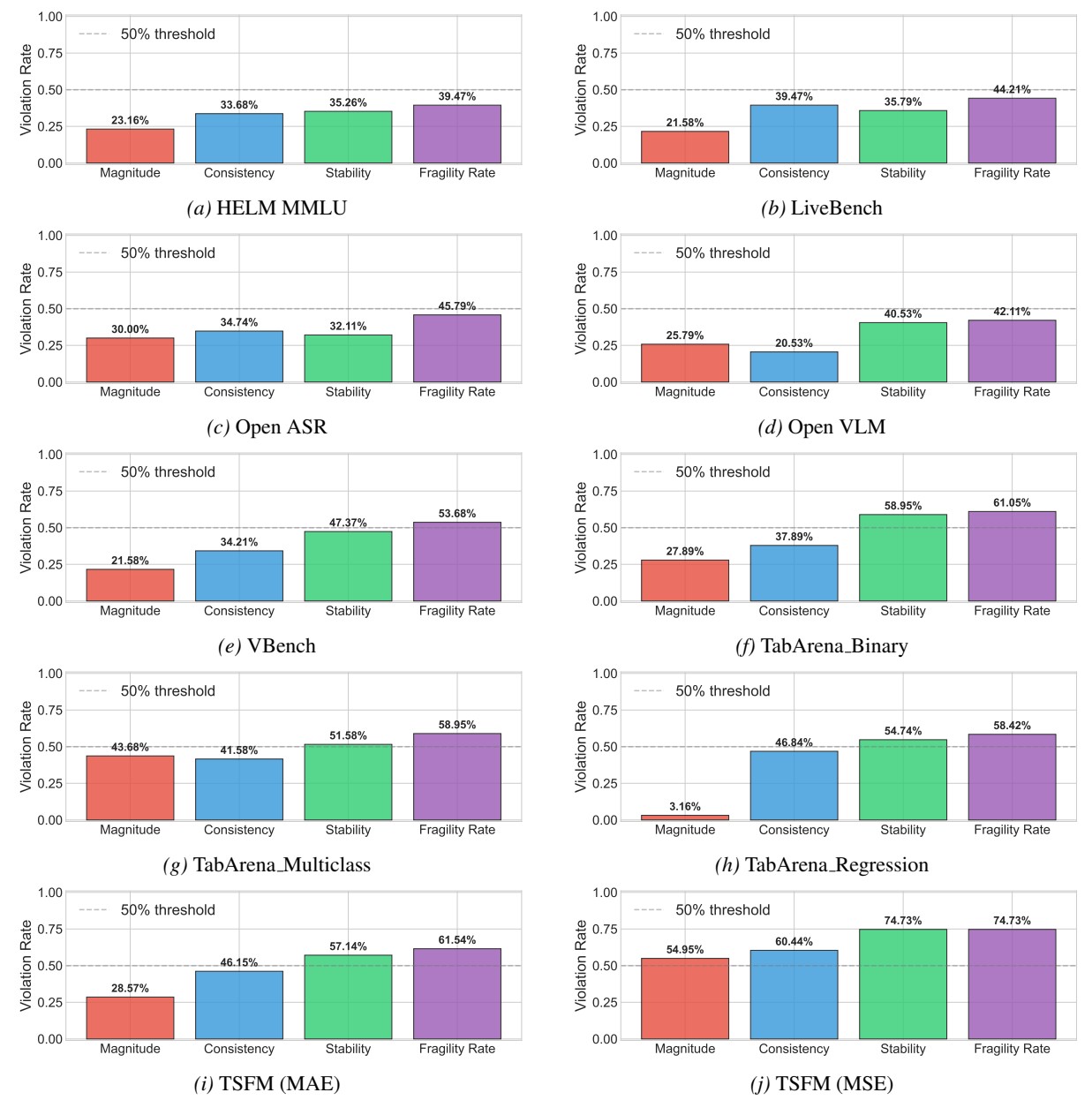

*Figure 6.* Violation rates by diagnostic test across ten benchmarks. The rightmost bar indicates the overall fragility rate.

*Table 5.* Fragility Rate by the number of top-ranked models.

| Number of Models | Violation Rates | | | Fragility Rate |
|---|---|---|---|---|
| | **Magnitude** | **Consistency** | **Stability** | |
| 10 | 35.56% | 40.00% | 48.89% | 53.33% |
| 20 | 23.16% | 33.68% | 35.26% | 39.47% |
| 30 | 22.99% | 28.28% | 35.86% | 38.85% |
| 40 | 19.10% | 24.62% | 30.13% | 32.56% |
| 50 | 14.61% | 19.10% | 23.35% | 25.22% |
| 60 | 12.03% | 15.65% | 18.93% | 20.45% |

*Table 6.* Fragility Rate by the number of sampled datasets.

| Number of Datasets | Violation Rates | | | Fragility Rate |
|---|---|---|---|---|
| | **Magnitude** | **Consistency** | **Stability** | |
| 7 | 9.47% | 30.00% | 14.21% | 30.00% |
| 17 | 12.11% | 30.53% | 24.74% | 35.26% |
| 27 | 20.00% | 33.68% | 31.05% | 40.53% |
| 37 | 18.95% | 31.05% | 32.11% | 35.79% |
| 47 | 21.58% | 32.63% | 34.74% | 37.89% |
| 57 | 23.16% | 33.68% | 35.26% | 39.47% |

Figure 6 shows the breakdown by magnitude, consistency, and stability along with the overall fragility rate. The dominant source of fragility varies by domain. Some benchmarks exhibit high consistency violations while others are constrained primarily by stability. This heterogeneity indicates that fragility arises from domain-specific characteristics rather than a

single systematic cause. Identifying which structural properties of a benchmark predict its dominant failure mode remains a direction for future work.

Furthermore, Tables 5 and 6 report the numerical values corresponding to Figure 4 using HELM MMLU. Table 5 shows that fragility concentrates among top-ranked models where SOTA claims are most common. Table 6 confirms that adding tasks does not resolve instability. This supports our argument that fragility reflects structural heterogeneity rather than an insufficient sample size. These findings reinforce our central argument. Fragility is most pronounced precisely where SOTA claims concentrate. Furthermore, methodological remedies such as adding tasks do not resolve this issue. Consequently, the appropriate response is not larger benchmarks but calibrated reporting.

## D. Stability Across Thresholds

### D.1. Single-Parameter Sensitivity

We examine the robustness of our findings regarding threshold selection across all benchmarks. Figures 7–16 illustrate how violation rates vary as each threshold changes while others remain at their default values ($\tau_w = 0.6$, $\tau_d = 0.2$, $\tau_b = 0.2$).

Across all benchmarks, the fragility rate remains stable as individual thresholds vary. This persists even though the violation rates for each test change substantially. This stability indicates that our findings do not depend on specific threshold choices. The dominant source of fragility differs by domain. Some benchmarks are constrained primarily by consistency while others are constrained by stability. However, the overall pattern persists regardless of which threshold is adjusted.

These patterns indicate that threshold selection does not drive our conclusions. The claim-evidence gap persists under lenient, moderate, and strict standards alike. This robustness supports treating fragility as a structural property of benchmark-based comparison rather than an artifact of our analytical choices. We emphasize that these thresholds operationalize measurement rather than define publication standards. While different thresholds yield varying violation rates, the qualitative findings remain unchanged. Thus, fragility is pervasive regardless of the specific cutoffs selected.

### D.2. Cross-Benchmark Consistency Relaxation

The single-parameter analysis above varies each threshold independently. Relaxing the consistency threshold to $\tau_w = 0.5$, which requires only a bare majority of task-level wins, reduces consistency violations in most benchmarks. Magnitude ($\tau_d = 0.2$) and stability ($\tau_b = 0.2$) thresholds remain unchanged.

*Table 7.* Effect of relaxing the consistency threshold from $\tau_w = 0.6$ to $\tau_w = 0.5$ across all ten benchmarks.

| $\tau_w$ | Diagnostic | HELM MMLU | LiveBench | Open ASR | Open VLM | VBench | TabArena _Binary | TabArena _Multiclass | TabArena _Regression | TSFM (MAE) | TSFM (MSE) |
|---|---|---|---|---|---|---|---|---|---|---|---|
| 0.6 | Consistency | 33.68% | 39.47% | 34.74% | 20.53% | 34.21% | 37.89% | 41.58% | 46.84% | 46.15% | 60.44% |
| | Fragility Rate | 39.47% | 44.21% | 45.79% | 42.11% | 53.68% | 61.05% | 58.95% | 58.42% | 61.54% | 74.73% |
| 0.5 | Consistency | 14.74% | 16.32% | 34.74% | 12.11% | 18.95% | 14.21% | 41.58% | 39.47% | 17.58% | 45.05% |
| | Fragility Rate | 35.26% | 36.32% | 45.79% | 40.53% | 48.42% | 58.95% | 58.95% | 55.26% | 57.14% | 74.73% |

Fragility rates decrease only slightly and remain between 35% and 75% in all cases. Some benchmarks show no or negligible change, which indicates that magnitude or stability violations drive their fragility. Even under this lenient threshold, the claim-evidence gap persists across all domains.

### D.3. Task/Dataset Heterogeneity Within Benchmark

Benchmark tasks vary in how homogeneous their evaluation units are. Some aggregate closely related datasets within a single task, while others combine distinct capabilities or opposing criteria. This heterogeneity affects diagnostic outcomes. A low win rate may reflect task conflict or weak performance. A small effect size may reflect heterogeneous scales or negligible improvement. A low breakdown point may reflect diversity in the benchmark or dependence on a few tasks.

Our diagnostics do not distinguish between these sources. They reveal whether aggregate claims align with task-level evidence. These sources relate to benchmark design, which we do not critique. When a benchmark is highly heterogeneous, aggregate SOTA claims become less informative. Reporting diagnostics alongside mean scores makes these patterns visible and allows readers to evaluate the evidence directly. Quantifying task heterogeneity and its relationship to ranking stability remains an open direction for community study.

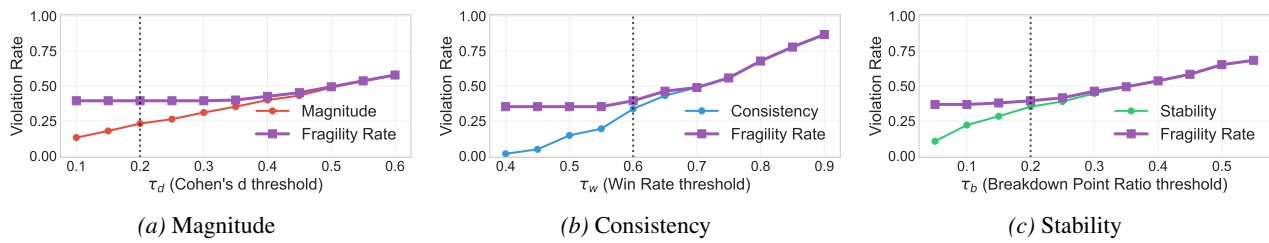

*(a)* Magnitude       *(b)* Consistency       *(c)* Stability

*Figure 7.* Sensitivity analysis of violation rates to threshold selection on the HELM MMLU leaderboard.

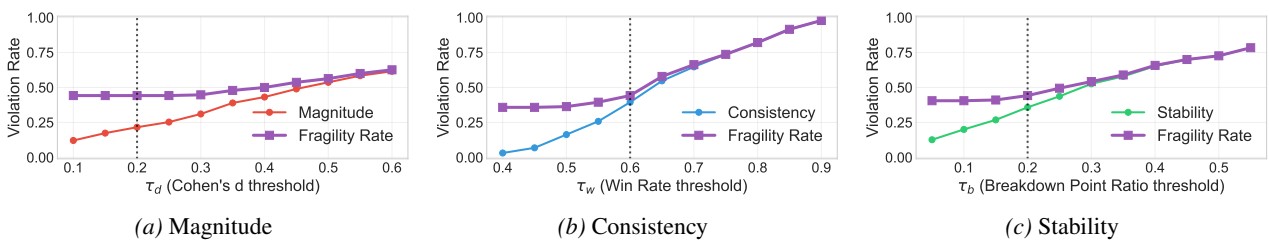

*(a)* Magnitude       *(b)* Consistency       *(c)* Stability

*Figure 8.* Sensitivity analysis of violation rates to threshold selection on the LiveBench leaderboard.

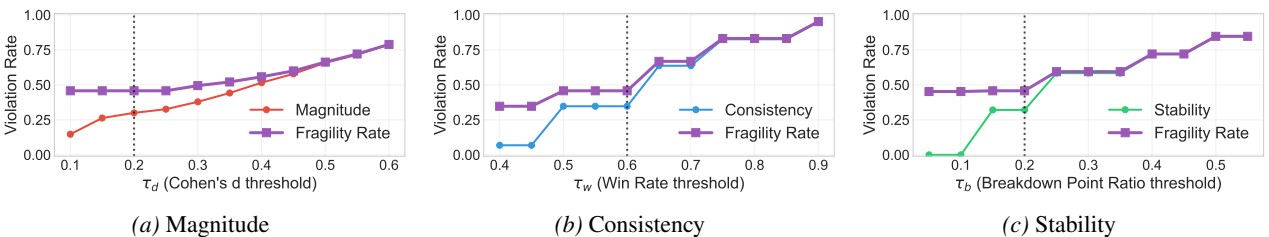

*(a)* Magnitude       *(b)* Consistency       *(c)* Stability

*Figure 9.* Sensitivity analysis of violation rates to threshold selection on the Open ASR leaderboard.

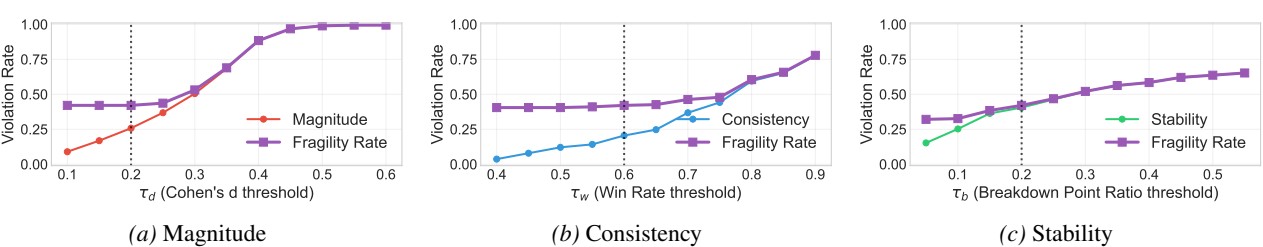

*(a)* Magnitude       *(b)* Consistency       *(c)* Stability

*Figure 10.* Sensitivity analysis of violation rates to threshold selection on the Open VLM leaderboard.

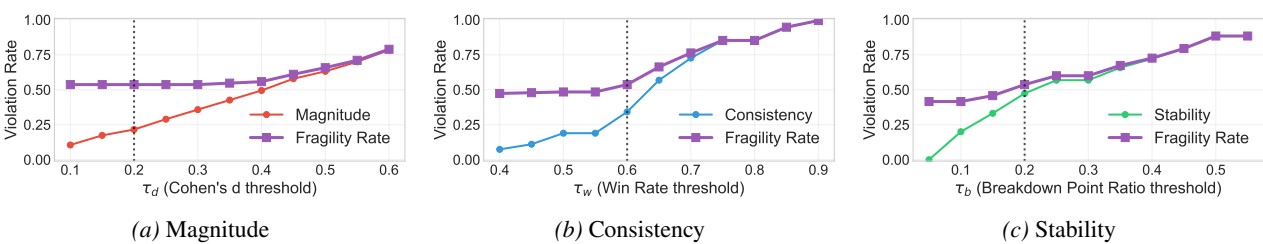

*(a)* Magnitude       *(b)* Consistency       *(c)* Stability

*Figure 11.* Sensitivity analysis of violation rates to threshold selection on the VBench leaderboard.

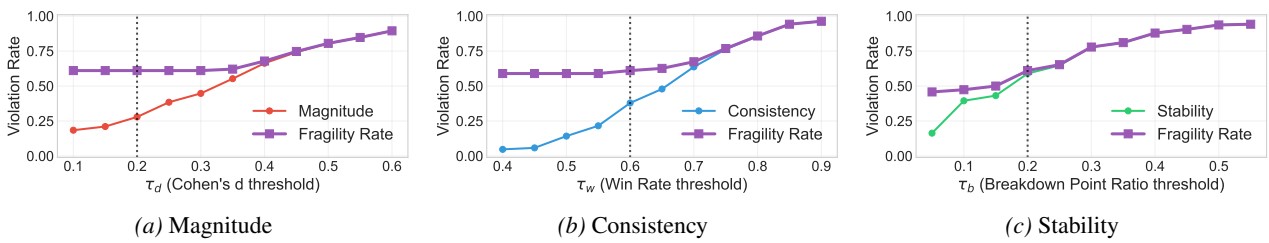

*(a)* Magnitude       *(b)* Consistency       *(c)* Stability

*Figure 12.* Sensitivity analysis of violation rates to threshold selection on the TabArena_Binary leaderboard.

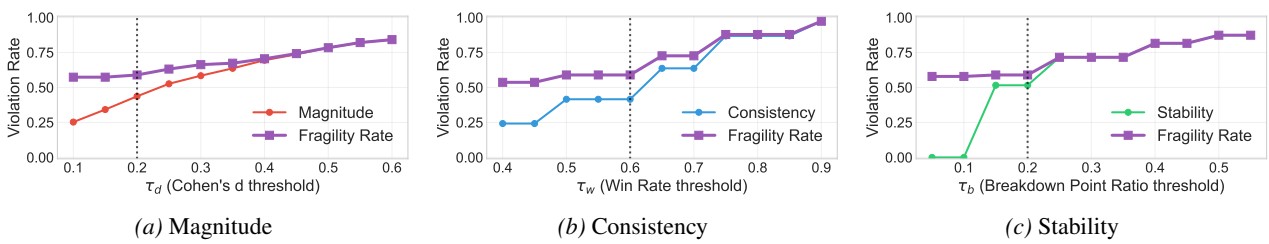

*(a)* Magnitude       *(b)* Consistency       *(c)* Stability

*Figure 13.* Sensitivity analysis of violation rates to threshold selection on the TabArena_Multiclass leaderboard.

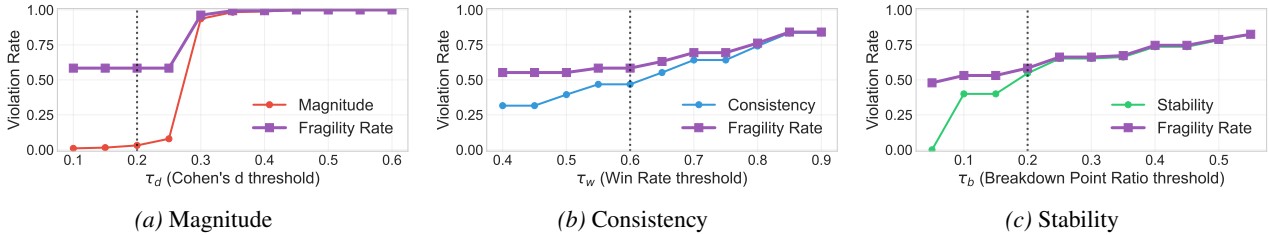

*(a)* Magnitude       *(b)* Consistency       *(c)* Stability

*Figure 14.* Sensitivity analysis of violation rates to threshold selection on the TabArena_Regression leaderboard.

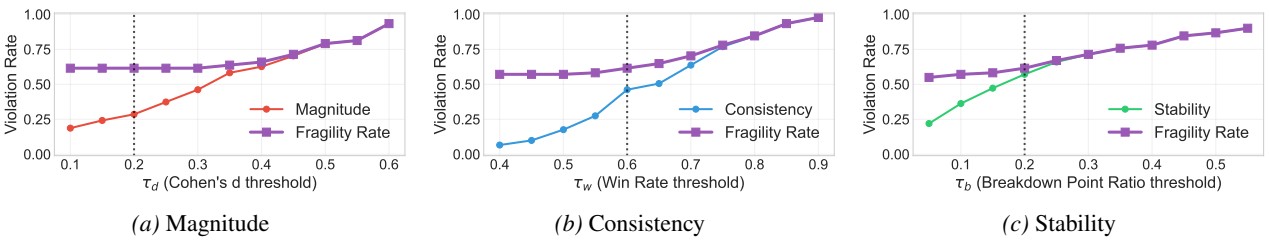

*(a)* Magnitude       *(b)* Consistency       *(c)* Stability

*Figure 15.* Sensitivity analysis of violation rates to threshold selection on the TSFM (MAE) leaderboard.

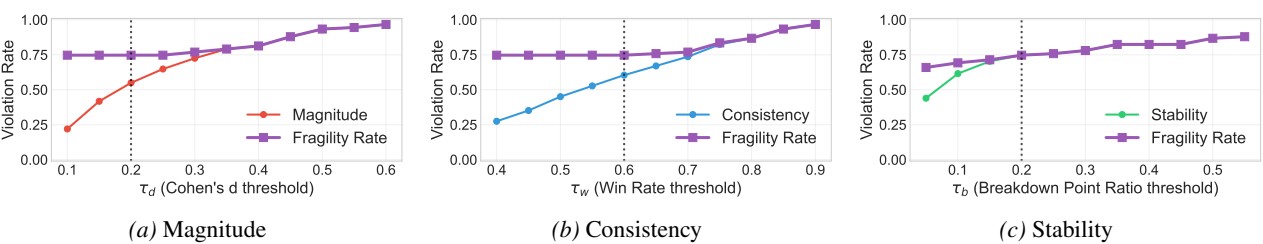

*(a)* Magnitude       *(b)* Consistency       *(c)* Stability

*Figure 16.* Sensitivity analysis of violation rates to threshold selection on the TSFM (MSE) leaderboard.

