# OpenReview forum: "Position: State-of-the-Art Claims Require State-of-the-Art Evidence"
_ICML.cc/2026/Position_Paper_Track — ICML 2026 Position Paper Track regular_

### Official Review · Reviewer_Ygzp · 2026-03-02

**Significance:** 4
**Argument Clarity:** 4
**Rating:** 6
**Confidence:** 4

**Questions:**

1. Did you collect any information about whether the results you collected were from papers published in ML conferences? Do you believe that your sample of benchmark results is representative of those from ML conferences?

A minor suggestion: To get an indicator symbol in Equations (4) and (6), you should use \mathds{1} from the dsfont package instead of \mathbb{1}, which creates a weird symbol instead.

**Alternative Views Section:**

Yes

**Compliance With Llm Reviewing Policy A Conservative:**

Affirmed.

**Discussion Potential:**

4

**Final Justification:**

I stand by my original (positive) evaluation of the paper: they address an important problem, provide strong quantitative evidence for their position rooted in robust statistics, and make realistic suggestions for the community to improve the issue. My only concern about the paper was whether the results they analyzed were representative of top-tier ML venues, and the authors have clarified the source and characteristics of this data in their rebuttal.

**Paper Summary:**

This paper argues that state claims of "state of the art" performance in ML papers often rely on insufficient evidence, and that more accurate language is needed for honest scientific communication. This argument is backed up through a quantitative analysis of the fragility of evidence for claims of superior performance on ML benchmarks: for ten public benchmarks with public evaluations, the authors analyze the magnitude, consistency, and stability of model comparisons. Overall, the analysis shows that even when one model outperforms another in terms of mean score across datasets, the superiority is often quite fragile along at least one of the tests (magnitude, consistency, and stability), particularly among models reported as top performing on a given benchmark. The paper concludes with suggestions for authors, reviewers, and venues, with the goal of modifying our community norms around accurate claims of algorithm superiority.

**Position:**

Yes

**Position In Title:**

Yes

**Related Work:**

4

**Strengths And Weaknesses:**

Strengths:
1. The issue is important. Many researchers are aware that our publication system has misaligned incentives which lead to a lot of questionable methodology.
2. The authors provide strong evidence for their position. They are doing the essential dirty work of empirical research that most don't want to do: understanding and cleaning up the mess made by the more mainstream task of proposing a new algorithm and claiming superior performance. This work involves a good amount of collecting empirical results from many different domains (vision, language, time series) and thorough analysis with a solid methodology rooted in robust statistics.
3. The authors make reasonable, realistic suggestions for the community to address the problem. They don't propose new evaluation methodology, but a change in the standards of our reviewers and venues around language used to claim superior performance.

Weaknesses:
1. My only concern is whether the benchmark results analyzed by the authors are a good representation of the benchmark results reported in ML conference papers. I couldn't find in the paper any description about whether the collected results were classified in terms of their publication status. It could be that some of these results are published in less popular venues, or not even published at all. By the way, I'm inclined to guess that the collected results are indeed representative of ML conferences, but with the information available it is hard to say for sure. Also, I can see why it could be difficult to automate the collection of this information.

**Support:**

4

---

> ### Author Rebuttal · Authors · 2026-03-29
>
> Thank you for the reviewer's comments. Reviewer Ygzp asks about the representativeness of our benchmark analysis and the source of collected results, which we clarify below.
>
> ## Representativeness of Benchmark Results
> We will include the following context for each benchmark in Section 4.3 and Appendix B in the revised version.
>
> > We selected benchmarks that are widely adopted and peer-reviewed. HELM MMLU (Hendrycks et al., 2021) is one of the most cited benchmarks in NLP and serves as a standard evaluation for large language models (LLMs). The remaining benchmarks were introduced at top-tier venues: LiveBench (White et al., 2025) at ICLR 2025 (Spotlight), VLMEvalKit (Duan et al., 2024) at ACM Multimedia 2024, VBench (Huang et al., 2024) at CVPR 2024 (Highlight), TabArena (Erickson et al., 2025) at NeurIPS 2025 Datasets and Benchmarks (Spotlight), and TSFM-Bench (Li et al., 2025) at KDD 2025. The Open ASR Leaderboard (Srivastav et al., 2025), while released most recently as a preprint, is included because it evaluates frontier speech models from major industry labs alongside open-source systems. This makes the benchmark relevant for practical comparison.
>
> We selected ten benchmarks that span diverse domains and remain actively maintained by their research teams as of 2025 (Table 1 reports the latest update dates). With the exception of MMLU, all benchmarks were published in 2024 or 2025 at top-tier venues, providing a curated and practically relevant setting for this study.
>
> The analysis does not question benchmark design. As Reviewer **zXmh** notes, our analysis applies specifically to multi-task aggregate benchmarks. Thus, the benchmarks selected reflect this scope and represent substantial community contributions. The focus is on how their results are interpreted.
>
>
> ## Publication Status of Analyzed Model Entries
> We did not classify individual model entries by publication status. The analysis uses the top 20 models by mean score on each leaderboard, without filtering for peer-reviewed publications.
> Top-ranked model entries are dominated by frontier models from well-resourced organizations with publications at NeurIPS, ICML, ICLR, and similar venues or journals. Also, these leaderboards aggregate results from models reported across multiple venues, and therefore reflect how evaluation is conducted in practice rather than a separate or non-representative setting. We will clarify these points in Appendix C.
>
>
> ## Latex Suggestion
> We will use '\mathds{1}' from the `dsfont` package in Equations (4) and (6) in the camera-ready version.
>
>
> ## Additional Corrections
> - We will clarify the caption of Figure 1, as suggested by Reviewer zXmh. We will also define the terminology and scope of the paper. Section 6 states these limitations.
>
> - There is a typo in Table 1. Vbench should be corrected to `VBench.' The total number of models in OpenVLM is 284, not 28.
>
> - We verified the number (Figure 5 and Appendix D) and corrected L#239:
> > However, the overall fragility rate remains above 30% across all single-parameter configurations.
>
> - Please refer to the response to Reviewer **zXmh**. We include revised concluding remarks that reflect all reviewer comments.
>
>
> ---
> We thank the reviewers for their careful reading and constructive feedback.

---

> > ### Author Rebuttal · Reviewer_Ygzp · 2026-04-01
> >
> > Thank you for your response. I think that the paper will be improved if you incorporate this information about the status of the benchmarks and model entries. All of my other concerns are resolved.

---

### Official Review · Reviewer_QTb3 · 2026-03-07

**Significance:** 4
**Argument Clarity:** 4
**Rating:** 5
**Confidence:** 4

**Questions:**

Please refer to the Strengths and Weaknesses section.

**Alternative Views Section:**

Yes

**Compliance With Llm Reviewing Policy A Conservative:**

Affirmed.

**Discussion Potential:**

4

**Final Justification:**

I thank the authors for responding to my comment. I maintain that this paper advances an important position that is essential to improving objectivity and transparency in the community. I understand that "tasks conflict is a property of benchmark design, not reporting practice", and this position is mainly about good reporting practices. My concern was the choice of task heterogeneity as a marker of SOTA - groups of tasks may be constructed where there is a uniform distribution of conflicting tasks, so no single model consistently wins more than the others - hence, no SOTA exists according to this definition. I get the general point that the more the number of tasks a model wins in, the better it is - but there could be a theoretical limit to this number (in the presence of conflict), which is better to clarify in the paper.

**Paper Summary:**

The authors observe "that state-of-the-art" (SOTA) performance claims in the machine learning / artificial intelligence community are often too strong, given the nature of the methods. They find that a large number of methods that make such a claim fail on at least of the three basic conditions that SOTA models ought to satisfy. The authors propose reframing claim language to be more faithful to reality as a simple initial solution to this problem.

**Position:**

Yes

**Position In Title:**

Yes

**Related Work:**

3

**Strengths And Weaknesses:**

## Strengths:

1. The authors highlight an important gap in the honesty with which state-of-the-art (SOTA) claims are made in ML / AI research. They observe that current claim languages are too strong and need to be aligned with the reality.

2. The authors perform several statistical tests and find that a large number of methods that are claimed to be SOTA fail in at least on of the basic criteria that SOTA models must satisfy. Since existing SOTA claims are based on simple, aggregate metrics such as mean error rate, their coarse-grained nature do not capture the nuances of a task that they ought to should a method become SOTA.

3. The authors propose simple first-step solutions based on honest reporting and revision of de facto claim language that require authors to be more transparent about and restrict themselves to just the true aspects over which superiorities are observed and nothing more. True SOTA claims should only be reserved for scenarios where a method passes a wide variety of statistical tests such as those proposed by the authors. The authors also prescribe simple, easy-to-adapt guidelines for both reviewers and venues involving parity checks between claim and evidence in submitted papers. As they rightly state, "This is the baseline expectation for honest scientific communication".



## Weaknesses:

1. While Effect Size is a well-established statistical test, it is important to provide more details about it so that readers, who are not aware of its specifics can understand its purpose and relevance to the position better. Specifically, an intuitive description of what quantity / aspect of the methods the Effect Size test measures and how it does that would help improve understandability and the utility of the test in the present context.

2. The proposed analysis of metric fragility reveals a crucial weaknesses in the standard evaluation protocols adopted community-wide. The authors of this position suggest reframing claim language as an initial solution to this issue. In a hypothetical world, assume such a change is actually effected - "state-of-the-art" is replaced by a more faithful description such as "ranks first on this benchmark". Now, due to the nature of the community to get fixated on some unitary criterion (such as reviewers routinely requesting SOTA comparisons, as observed by the authors), all that the success of this position could result in is possibly the above substitution of terminology and nothing more. In other words, due to subconscious biases, "ranks first on this benchmark" becomes the new "state-of-the-art".  The core issue of proxies diverging from true objectives remain unsolved. For this reason, it might be worth taking a stronger position advocating for the holistic reporting of a method's statisical evalutions - such as all the three statistical tests, while reporting its performance. A method need not pass all the tests but having them in the public domain providing a more transparent and complete picture.

3. Although the authors illustrate that a signifant number of existing approaches that claim SOTA do not qualify as such under their tests and propose a revision of claim language to align description and reality, they do not really enumerate, perhaps to the fullest extent, what could comprise SOTA. The considered tests seem to be a subset of necessary conditions that methods must satisfy in order to be SOTA. But are they all strictly necessary? Is this subset complete or should researchers consider a more complete suite of tests? And finally, who is the authority to determine what actually comprises SOTA? Is it something that should be arrived at throught consensus at the community-level? Given the number of aspects of evalution that need to be taken into account, arriving at such a consensus seems like a matter of very long time and lots of deliberation.

**Support:**

4

---

> ### Author Rebuttal · Authors · 2026-03-29
>
> Thank you for the reviewer's comments. Reviewer QTb3 raises several detailed points regarding interpretation and presentation, which we address below.
>
>
> ## Intuitive Description of Effect Size
>
> The reviewer notes that readers unfamiliar with Cohen's $d$ would benefit from an intuitive explanation. We agree and will add the following to Section 3.3 in the camera-ready.
>
> > Cohen's $d$ compares the performance gap between two models to the variation of their scores across tasks. A value of $d = 0.2$ means the gap only slightly exceeds task-to-task variation. The default $\tau_d = 0.2$ therefore sets a small effect threshold. Failure indicates that the gap is negligible relative to cross-task variance. Two models may differ in mean yet remain practically indistinguishable.
>
>
> ## "Ranks First" May Become the New "SOTA"
>
> The reviewer identifies that changing terminology alone may transfer bias to a new label. We agree. This concern aligns with our discussion in Section 5.1, and we will adopt the phrase "subconscious biases" in the revised version.
>
> The Call to Action requires care. Many venues include statistical testing in their checklists, but these requirements rarely specify the depth of analysis. No single stakeholder can determine this unilaterally. Authors, reviewers, and venues must each reconsider their role (as discussed in Section 5.4).
>
> We will further refine Section 7 and the concluding remarks to the call to action (See the response to Reviewer **zXmh**).
> Therefore, we argue:
>
> > L#70. Individual authors cannot shift equilibria alone. However, venue-level guidelines can facilitate this shift.
>
> The three diagnostics move the discussion from labels to evidence. When win rates, effect sizes, and stability are reported alongside aggregate scores, readers can evaluate the underlying evidence directly rather than rely on a label. This aligns with the reviewer's suggestion for holistic reporting and reduces the risk of label substitution. Specifically, we will revise Section 7. For instance, revised Section 7.1 (for author) as follows:
>
> > We do not propose that authors must pass specific statistical tests. Instead, we propose that claim language should reflect what the evidence actually supports. When aggregate gains are narrow or inconsistent, phrases like "achieves lowest average error" or "ranks first on this benchmark" are more accurate than "state-of-the-art," which serves as a marketing term rather than a precise scientific claim. However, terminology revision alone is insufficient if "ranks first" inherits the same subconscious connotations. Accompanying such claims with win rates and effect sizes prevents any single label from serving as an uncritical proxy for superiority.
>
>
> ## Completeness of the Three Tests in Section 3.2
>
> Several expressions in the submitted version were more assertive than intended—in particular the checkmark list in Section 3.2.
>
> > L#157. We do not argue that every criterion must be satisfied. We argue that when any of these elementary tests fails, the comparison is fragile, and the strength of the claim should be tempered accordingly.
>
> The tests target consistency, magnitude, and stability, which are commonly assumed when SOTA is claimed. Other properties such as efficiency, fairness, and calibration may also matter depending on the application. We do not present a complete definition of SOTA. We show that even minimal tests reveal widespread fragility. We will also correct misleading points. For example, we will revise the checkmarks in Section 3.2 to avoid misinterpretation.
>
>
> ## Authority
>
> Defining SOTA should not rest with a single paper. Meaningful superiority is determined by community norms, including venue standards and reviewer expectations, similar to how clinical fields established reporting guidelines, which is discussed in Section 5.4. This work makes implicit assumptions explicit and testable, and provides a basis for discussion.
>
>
> ## Additional Corrections
> - As suggested by the Reviewer **Ygzp**, we will use \mathds{1} from the dsfont package in Equations (4) and (6) in the camera-ready version.
>
> - We will clarify the caption of Figure 1, as suggested by Reviewer zXmh. We will also define the terminology and scope of the paper. Section 6 states these limitations.
>
> - There is a typo in Table 1. Vbench should be corrected to `VBench.' The total number of models in OpenVLM is 284, not 28.
>
> - We verified the number (Figure 5 and Appendix D) and corrected L#239:
> > However, the overall fragility rate remains above 30% across all single-parameter configurations.
>
> - Please refer to the response to Reviewer **zXmh**. We include revised concluding remarks that reflect all reviewer comments.
>
>
> ---
> We thank the reviewers for their careful reading and constructive feedback.

---

> > ### Author Rebuttal · Reviewer_QTb3 · 2026-04-04
> >
> > I thank the authors for addressing my comments. All of my concerns have been resolved and I maintain that the paper advances an important position. The only challenge (perhaps not for the position but that for the field if the proposed approaches were to be adopted) would be arriving at a community-wide consensus of what metrics to value when evaluating a work. There is nothing concrete in this position that provides guidance in this regard (it may not even be possible to provide unified guidance since it could be task-dependent, but there is no discussion on this either). It is not so much a criticism of this position, perhaps even an endorsement, since it makes an existing limitation explicit - adoption of the proposed actions would introduce (hopefully) a transient period of turbulence in the community where lots of works would be misevaluated simply due to the lack of consensus on what is valuable, before a settlement on some unified standard.
> >
> > Reviewer nBjk makes some valid points regarding task conflicts, where the authors require SOTA models to consistently win across tasks. This argument hints at what is described by the No Free Lunch Theorem, pointing to scenarios where there are no clear winners when a large enough variety of tasks is considered. This is not in disagreement with what the authors try to advance through this position (they never mention methods ought to excel in all possible tasks, but only to not appear to be mentioning anything to that effect when they in fact do not), but only brings into question the choice of this criterion of "consistent wins" as a characteristic for defining SOTA. However, the authors mention fragile performance in claimed SOTA models occur even in the absence of conflict, which is a valid argument - so it is worth making this detail clear, i.e., that the authors mean to refer to consistent wins across tasks that are not conflicting in nature.

---

### Official Review · Reviewer_zXmh · 2026-03-11

**Significance:** 4
**Argument Clarity:** 4
**Rating:** 6
**Confidence:** 5

**Questions:**

No pressing questions

**Alternative Views Section:**

Yes

**Compliance With Llm Reviewing Policy A Conservative:**

Affirmed.

**Discussion Potential:**

4

**Paper Summary:**

This paper proposes to enforce stricter standards for claiming state of the art (SOTA) performance in Machine Learning benchmarks that consist of multiple tasks.
The authors first identify the rising number of papers claiming sota, and the reduction of the evidence backing these claims to just being the best on average score. The paper proposes a wider range of reasonable metrics that should be statisfied to back SOTA claims and demonstrates that across popular benchmarks, SOTA claims are fragile since they violate at least 1 of the introduced `superiority metrics`.

**Position:**

Yes

**Position In Title:**

Yes

**Related Work:**

3

**Strengths And Weaknesses:**

*Strengths*
1. The alternative views section is quite comprehensive and each of the rebuttals are reasonably convincing
2. The paper's position does not require an overhaul to existing experimental practice but rather imposes guidance on tighter language for claims made in papers.
3. The paper buttresses it's claims with showing a comprehensive list of benchmarks on which sota claims are fragile
4. Paper provides recommended actions for different sets of actors -- reviewers, authors and conference organizers

*Weaknesses*
1. The paper only applies to benchmarks that feature a  set of tasks -- mentions but does not address the single task SOTA setting.
2. Whilst figure 1 on the surface provides a good motivation, the graph is confounded by SOTA mentions w.r.t single task settings, which their concerns / methodology does not cover

**Support:**

4

---

> ### Author Rebuttal · Authors · 2026-03-29
>
> Thank you for the reviewer's comments. Reviewer zXmh highlights scope limitations and terminology issues, which we clarify below.
>
>
> ## Scope and Terminology
>
> The reviewer notes that the analysis applies to benchmarks with multiple evaluation units and does not cover the single-task setting. We agree and clarify this scope in the revised Appendix.
>
> > Our framework treats these uniformly as evaluation units $d \in \mathcal{D}$ over which aggregate scores are computed. The mathematical structure, ranking models by mean score across $k$ units, is identical regardless of the unit definition.
>
> We will add a short paragraph in the appendix to clarify how evaluation units are defined across domains. The claim-evidence gap arises from aggregation across these units. In single-task settings, fragility arises from different sources (e.g., test variance), which are outside our current scope.
>
> The Limitations section (L#430) states that single-metric focus and multi-metric evaluation define the scope boundaries. Extending the framework to these settings remains future work.
>
>
> ## Figure 1 Caption and Explanation
> The reviewer observes that Figure 1 includes SOTA mentions from both single-task and multi-task settings, which conflates motivation with the empirical scope. We will revise the caption to make this distinction explicit:
>
> > Published papers mentioning ``state-of-the-art'' in abstracts across major AI conferences (2021--2025). Darker bars indicate SOTA mentions, and lighter bars indicate other papers. In this study, the empirical analysis in Section 4 focuses on benchmarks that aggregate across multiple tasks or datasets.
>
>
> ## Additional Corrections
> - As suggested by the Reviewer **Ygzp**, we will use \mathds{1} from the dsfont package in Equations (4) and (6) in the camera-ready version.
>
> - There is a typo in Table 1. Vbench should be corrected to `VBench.' The total number of models in OpenVLM is 284, not 28.
>
> - We verified the number (Figure 5 and Appendix D) and corrected L#239:
> > However, the overall fragility rate remains above 30% across all single-parameter configurations.
>
> ---
>
> ## Updated Concluding Remarks
> Due to space limits, we summarize the revised concluding remarks here. Based on the all reviewers' feedback, we will include the following in the revised version.
>
> > **Matching Claims to Evidence.**
> > Our goal is to align claim language with the strength of the supporting evidence. In current practice, aggregate rankings are often interpreted as broad superiority, even when the underlying evidence may be narrow or heterogeneous. Reporting language should reflect these limitations, especially when performance varies across tasks or depends on specific evaluation choices.
>
> > **Scope and Future Work.**
> > Our analysis focuses on single-metric, multitask benchmarks at a fixed snapshot in time. We do not address single-task settings or multi-metric decision scenarios, where different notions of superiority may apply. Future work includes extending these diagnostics to multi-metric evaluation, tracking fragility over time, and analyzing domain-specific patterns that lead to instability, as well as studying how reviewer–author interactions shape SOTA claims in practice.
>
> > **Institutional Context.**
> > The patterns we observe arise from widely adopted evaluation practices rather than isolated cases. Because these practices are shared across the community, changes in reporting are unlikely to occur at the level of individual papers alone. Aligning claims with evidence shifts attention away from marginal ranking differences and toward more stable comparisons.
>
>
> ---
> We thank the reviewers for their careful reading and constructive feedback.

---

> > ### Author Rebuttal · Reviewer_zXmh · 2026-04-01
> >
> > Gave a strong accept !

---

### Official Review · Reviewer_nBjk · 2026-03-23

**Significance:** 2
**Argument Clarity:** 2
**Rating:** 3
**Confidence:** 4

**Questions:**

In your "3.2. Deconstructing SOTA Claims", you claim that a paper claiming SOTA often implicitly claims that "The model wins consistently across tasks.". I agree with your other two points of having sufficiently large improvement and the ranking being stable across minor perturbations, but I don't agree with this  "wins across tasks" claim. Taking your "chemistry tutoring based on aggregate MMLU scores" example, if a model performs badly on the chemistry portion of the benchmark but performs so well on the other portions that it has better aggregate performance, then I think it still deserves to be called as a SOTA. It might be the case that the benchmark itself is poorly designed. In a hypothetical scenario, suppose a model which becomes better at the high_school_chemistry task in MMLU will always perform worse on the world_religions task. Then models might never become "SOTA" using your definition (for a benchmark which might have conflicting tasks).

**Alternative Views Section:**

Yes

**Compliance With Llm Reviewing Policy A Conservative:**

Affirmed.

**Discussion Potential:**

2

**Final Justification:**

I have increased my rating from 2: Reject to 3: Borderline reject in light of the revisions the authors claim they will incorporate (in the rebuttals to my review and also to the other reviewers). The authors mentioned in the rebuttal that they will tone down their claims, which partially alleviates my concerns. They also said they will clarify that fragility remains high even within homogeneous subtasks for some datasets they consider. Even after these points are addressed, I am still unable to understand why the other reviewers feel that the "Discussion Potential" for this paper is high. Unfortunately, I have to disagree with them about this.

**Paper Summary:**

The paper states that the term "state-of-the-art" has been overused by ML papers without sufficient backing evidence. They advocate that most "SOTA claims" should be tempered and replaced with more modest claims like "ranks first on this benchmark". They state that "SOTA performance" is used in papers to refer to an improvement in average performance over certain benchmarks. But that when a model is claimed to be SOTA, there is an implicit assumption that certain additional properties are also satisfied such as: the improvement should be meaningfully large, the model should win consistently on different subsets of the benchmark dataset, and that the model should still be superior even if the benchmark dataset is subject to minor perturbations. The paper empirically shows how several existing "SOTA" claims no longer remain SOTA if they are expected to satisfy all these three properties (which are explicitly defined by them in a reasonable way).

**Position:**

Yes

**Position In Title:**

No

**Related Work:**

3

**Strengths And Weaknesses:**

**Well-supported**: Yes, it is well-supported.

**Relevance**: Yes, the topic is relevant.

**Discussion potential**: I don't believe this particular paper will inspire much further discussion in the community. The issues pointed by the paper are not exactly new. They do cite several related works and also do state as one of their alternative views that "The Community Already Knows This". The paper has a problem with how ML papers use the SOTA term. I agree with them but also as they seem to agree, as a researcher becomes more experienced, they eventually don't attribute too much to this term. Interestingly, I was looking at the Wikipedia article on *State of the art*: https://en.wikipedia.org/wiki/State_of_the_art and it turns out that the broader issues which this paper has with the term SOTA have already surfaced in the advertising community decades ago. "A 1994 essay listed it among "the same old tired clichés" that should be avoided in advertising": https://www.google.co.in/books/edition/Management_from_A_to_Zweig/pxk6xGMjdbkC?hl=en&gbpv=1&pg=PA115&printsec=frontcover, Zweig, Mark C. (2010) [11 July 1994]. "Better Writing". Management from A to Zweig: The Complete Works of Mark Zweig. Fayetteville, Arkansas: ZweigWhite. p. 115. ISBN 978-1-60950-017-7.

**Argument clarity**: Yes, the paper is clearly argued but it somehow feels very hollow. I don't know how exactly to describe it. It felt like the same points are repeated again and again, but there is not enough "meat" in the paper.

**Related works**: Yes, I do believe that the paper has done a decent job with the related works. But I am not entirely sure as I am not an expert on this topic.

**Support:**

2

---

> ### Author Rebuttal · Authors · 2026-03-29
>
> Thank you for the reviewer's comments. Reviewer nBjk raises several points about clarity in our framing, which we address below.
>
> ## The Issues Are Not Exactly New / Same Points Repeated
>
> We do not define what SOTA must mean or require satisfying all criteria. We examine whether the evidence used to support SOTA claims holds under minimal diagnostics.
> Benchmark fragility has been discussed before, and Section 5.3 ("The Community Already Knows This") addresses this point. The observation that experienced researchers often discount SOTA claims is consistent with our position.
>
> Individual awareness has not corrected the problem. Over 30% of papers at major venues still claim SOTA in their abstracts (Figure 1), and practices recommended by Demšar (2006) [1] and Bouthillier et al. (2021) [2] remain underused.
>
> *[1] Demšar, J. (2006). Statistical Comparisons of Classifiers over Multiple Data Sets. Journal of Machine Learning Research, 7(1), 1–30.*
>
> *[2] Bouthillier, X., Delaunay, P., Bronzi, M., Trofimov, A., Nichyporuk, B., Szeto, J., Sepahvand, N. M., Raff, E., Madan, K., Voleti, V., et al. (2021). Accounting for Variance in Machine Learning Benchmarks. In Proceedings of Machine Learning and Systems (Vol. 3, pp. 747–769).*
>
> Reviewer **QTb3** highlights a related concern regarding "subconscious biases," and Reviewer **Ygzp** characterizes this work as "the essential dirty work of empirical research that most don't want to do." Both observations point to a gap between individual awareness and community practice. We will also strengthen the call to action, as summarized in the response to Reviewer **zXmh**.
>
> Our contribution is the cross-domain quantification of the gap between claim language and statistical evidence across ten benchmarks, six domains, and more than 1000 pairwise comparisons. These results show that individual awareness alone does not change current practice.
>
> ## "Wins Across Tasks" Disagreement
>
> Several expressions in the submitted version were stronger than we intended.
> The checkmark list in Section 3.2 and the term "overclaiming" in Section 3.4 read as definitional rather than diagnostic. We have revised these to clarify our intent. Furthermore, we have also revised the contribution statement (L#62):
>
> > "However, claiming 'state-of-the-art' carries implicit assumptions beyond mean score superiority. Our diagnostics make these assumptions explicit and testable."
>
> In the submitted version, this diagnostic intent is consistent with statements already present in the manuscript:
>
> > L#157. "We do not argue that every criterion must be satisfied. We argue that when any of these elementary tests fails, the comparison is fragile, and the strength of the claim should be tempered accordingly."
>
> The reviewer's chemistry-versus-religion example raises a valid point about task heterogeneity. But the "conflicting tasks" argument does not explain all fragility. TSFM-Bench evaluates a single task (time series forecasting), across datasets with no conflicting task types, yet shows the highest fragility in our analysis. This result indicates that aggregation contributes beyond task conflict. Roque et al. (2025) report a similar pattern, showing that dataset selection alone can make many methods appear "best in class" (Section 5.1).
>
> *[3] Roque, L., Cerqueira, V., Soares, C., & Torgo, L. (2025). Cherry-Picking in Time Series Forecasting: How to Select Datasets to Make Your Model Shine. Proceedings of the AAAI Conference on Artificial Intelligence, 39(19), 20192–20199.*
>
> Appendix D shows that fragility remains high under more lenient thresholds (35-75% with $\tau_w=0.5$). The recommendation does not require models to win every task. This suggests that reporting where models win can provide a clearer view of task-level performance. Reporting win rates alongside mean scores makes task-level performance visible at no additional cost.
>
>
> ## Summary of Revisions
>
> In response to this reviewer's feedback:
>
> - Revised the checklist in Section 3.2 to present the three properties as commonly held assumptions rather than definitions
> - Replaced "overclaiming" with neutral language in Section 3.4
> - Revised Section 7.1: "We do not require authors to pass specific statistical tests. Claim language should reflect what the evidence supports."
> - Added a terminology paragraph clarifying that the framework applies uniformly across task, dataset, and dimension structures
> - Please refer to the response to Reviewer **zXmh**. We include revised concluding remarks that reflect all reviewer comments.
>
>
> ---
> We thank the reviewers for their careful reading and constructive feedback.

---

> > ### Author Rebuttal · Reviewer_nBjk · 2026-04-04
> >
> > My main concerns regarding the "Discussion Potential" and "Argument Clarity" are hard to address in a rebuttal. I understand that other reviewers have a differing opinion but I would like to maintain my position. Thank you very much.

---

### Decision · Program_Chairs · 2026-04-30

**Decision:**

Accept (regular)

**Comment:**

There are compelling arguments provided both for and against this position. The primary concern against is related to the alternative view that the use of SOTA, and similar language, is partially the result of review processes that overindex on its importance. In other words, while the position itself may be valid, it has become an entrenched behavior that will not be changed regardless of how compelling a position may be presented. There is substantial disagreement about whether this will result in discussion because of broader community acknowledgement of the issue already existing. A primary reason to consider is that this work presents a grounded reflection on the language used to communicate scientific advances. This feels unusually timely given how scientific communication has been increasing challenged in so many parts of the world. Overall, this seems like a position that would be of interest to the ICML community, even if it stands primarily to make explicit something that many know already.